# Domain2Vec: Vectorizing Datasets to Find the Optimal Data Mixture without Training

**Mozhi Zhang** [1]  **Howe Tissue** ✉  **Lu Wang** [2]  **Xipeng Qiu** [1]

## Abstract

We introduce DOMAIN2VEC, a novel approach that decomposes any dataset into a linear combination of several *meta-domains*, a new concept designed to capture the key underlying features of datasets. DOMAIN2VEC maintains a vocabulary of meta-domains and uses a classifier to decompose any given dataset into a domain vector that corresponds to a distribution over this vocabulary. These domain vectors enable the identification of optimal data mixture for language model (LM) pretraining in a training-free manner under the *Distribution Alignment Assumption* ($\text{DA}^2$), which suggests that when the data distribution of the training set and the validation set is more aligned, a lower validation loss is achieved. Moreover, DOMAIN2VEC can be seamlessly integrated into previous works to model the relationship between domain vectors and LM performance, greatly enhancing the efficiency and scalability of previous methods. Extensive experiments demonstrate that DOMAIN2VEC helps find the data mixture that enhances downstream task performance with minimal computational overhead. Specifically, DOMAIN2VEC achieves the same validation loss on Pile-CC using only $51.5\%$ of the compute required when training on the original mixture of The Pile Dataset. Under equivalent compute budget, DOMAIN2VEC improves downstream performance by an average of $2.83\%$.

## 1. Introduction

Through training on large-scale text corpora, Large Language Models (LLMs) have demonstrated remarkable generalization capabilities (Touvron et al., 2023; OpenAI, 2024;

---

[1]School of Computer Science, Fudan University, Shanghai, China [2]Ritzz-AI. Correspondence to: Howe Tissue (project lead) <h-sun20@tsinghua.org.cn>.

*Proceedings of the 42nd International Conference on Machine Learning*, Vancouver, Canada. PMLR 267, 2025. Copyright 2025 by the author(s).

Qwen Team, 2024; DeepSeek-AI, 2024). The training datasets for LLMs are typically composed of multiple domains, each derived from different sources. Recent research has shown that the mixture proportions of these domains (named as the data mixture) can significantly influence the effectiveness of LMs (Hoffmann et al., 2022b; Xie et al., 2023b), with data from one domain potentially affecting the performance in others (Guo et al., 2022). Typically, the data mixtures used for training LLMs are often determined heuristically or based on downstream performance metrics. However, these methods are not scalable and always result in a suboptimal data mixture. Thus, identifying the optimal data mixture in a scalable and efficient manner remains a critical and challenging research question.

Recently, researchers have proposed various methods to predict the optimal data mixture. The first line of prior works implicitly adjusts the data mixture by selecting high-quality data from different domains or datasets (Lin et al., 2024; Ankner et al., 2024; Thakkar et al., 2023). The second line of work focuses on modeling the relationship between the data mixture and the performance of LLMs, and explicitly adjusts the data mixture across different datasets (Rae et al., 2022; Xie et al., 2023a; Sagawa* et al., 2020; Fan et al., 2023; Ye et al., 2024; Ge et al., 2024; Gu et al., 2024a; Que et al., 2024). While prior work has shown promising results, there are some key issues: **1) Computational Efficiency**: For example, although the proxy model in DoReMi (Xie et al., 2023a) has only 280M parameters, its estimated FLOPs are high to $3.7 \times 10^{19}$ for calculating only 22 datasets. Moreover, The computational complexity of these methods will grow non-linearly as the number of datasets increases. **2) Lack of Scalability**: After establishing the functional relationship between data mixtures and model performance (Ye et al., 2024; Liu et al., 2024), if the dataset composition changes (e.g., by adding new datasets or filtering low-quality data, etc), previously fitted functions cannot be directly applied. This requires resampling new data mixtures, retraining proxy models, and refitting the functions, severely limiting the scalability of these methods.

To address these issues, we introduce DOMAIN2VEC, a novel framework designed to vectorize datasets. This enables us to perform all operations for computing optimal

mixing ratios in domain vector space, which has broad applicability when datasets change. Specifically, DOMAIN2VEC maintains a vocabulary of meta-domains, and we hypothesize that *any dataset can be approximated as a linear combination of several meta-domains with a specific distribution.* This distribution could serve as the vector representation (domain vector) of a given dataset.

To efficiently identify the meta-domain composition of any given dataset, we propose to use a meta-domain classifier to generate the corresponding domain vector. Building upon DOMAIN2VEC, we introduce the ***Distribution Alignment Assumption*** (DA$^2$) to find optimal data mixtures for LM pretraining. DA$^2$ states that *lower validation loss can be achieved when the domain vector of the training dataset better aligns with the domain vector of the validation dataset.* Based on DA$^2$, we can easily find the optimal data mixture without training.

Moreover, DOMAIN2VEC can be seamlessly integrated into prior works like RegMix (Liu et al., 2024). Unlike previous methods that model the relationship between data mixtures and language model performance (Liu et al., 2024; Ye et al., 2024), we model the relationship between domain vectors provided by DOMAIN2VEC and model performance, further enhancing efficiency and scalability of previous works.

In summary, we highlight our contributions as follows:

1. We introduce DOMAIN2VEC to vectorize datasets and propose viewing datasets as combinations of meta-domains. We present an efficient pipeline for vectorizing datasets using a meta-domain classifier.

2. We propose the ***Distribution Alignment Assumption*** (DA$^2$), a training-free method for identifying the optimal data mixture. We further demonstrate how DOMAIN2VEC can be seamlessly integrated into prior work to improve efficiency and scalability.

3. We validate the effectiveness of DOMAIN2VEC+DA$^2$ and +RegMix in text generation and downstream tasks. Experimental results show that our method can accurately predict the performance of various data mixtures without training proxy models. Moreover, we can identify data mixtures that achieve downstream performance comparable to DoReMi (Xie et al., 2023a), using only 0.26% of its computational cost.

## 2. Domain2Vec

In this section, we introduce DOMAIN2VEC, an algorithm that decomposes a dataset into a linear combination of various meta-domains and allows us to represent underlying features of datasets through a normalized vector. We also outline a pipeline for constructing the vocabulary of DOMAIN2VEC and training a meta-domain classifier.

**Key Assumption.** DOMAIN2VEC maintains a vocabulary, a set of meta-domains. Assume we have $n$ meta-domains $\mathcal{D}_j^*$ ($1 \leq j \leq n$), where $\mathcal{D}_j^*$ is represented as $e_j$, a one-hot vector where only the $j$-th element is 1. We hypothesize that, for any given dataset $\mathcal{D}$, it could be represented as a domain vector $\boldsymbol{v}$, by linear combination of these meta-domains. Specifically,

$$\boldsymbol{v} \approx \sum_{j=1}^{n} v_j \cdot e_j, \tag{1}$$

where each element $v_j$ of $\boldsymbol{v}$ represents the projection (weight) of the dataset $\mathcal{D}$ on $\mathcal{D}_j^*$. Thus, $\boldsymbol{v} = [v_1, v_2, ..., v_n]^\top$ can be a representation (distribution) of the dataset $\mathcal{D}$ over the meta-domains. However, an ideal approach for constructing these meta-domains remains to be established. Next, we will introduce how we construct meta-domains from large-scale unlabeled text corpora.

**Constructing the Vocabulary of DOMAIN2VEC.** With the above key assumption, we define meta-domains as a collection of actual datasets (or a set of domains) that serve as a *basis* in the domain vector space, allowing for linear combinations of these concrete datasets to represent any unknown domain in this space. These constructed meta-domains, which could represent datasets from any source, should satisfy the following three properties, similar to the properties of a basis in linear algebra:

1. **Spanning Set.** The domains that compose meta-domains should be as diverse and comprehensive as possible.

2. **Linear Independence.** There should be distinct differences between these constructed meta-domains.

3. **Computational Efficiency (Optional).** The method for constructing meta-domains should be computationally efficient.

For diverse and comprehensive meta-domains, we collect data from more than 100 coarse sources across English, Chinese[1], and Code. After deduplication, we obtain around 5.2 **TB** text data including more than 1 billion documents. The large corpora have a similar source composition as the standard large-scale LLM pretraining, including common crawl (CC), Wikipedia, social media platform, arXiv, code, news, books, etc. One could assume that the corpora already include as diverse and comprehensive contents as possible, corresponding to the requirement "spanning set" [2].

---

[1] In this paper, we primarily aim at languages of English and Chinese.

[2] Due to deduplication pre-processing and the native difference among the corpora, the requirement "linear independence" is also naturally satisfied.

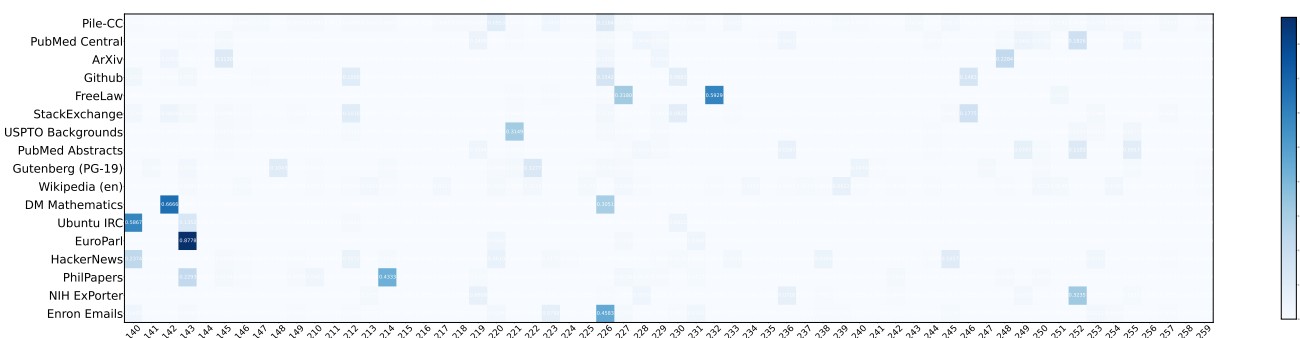

*Figure 1.* The domain vector of each sub-dataset of The Pile (Gao et al., 2021), where each row corresponds to a sub-dataset and each column corresponds to a meta-domain. The higher the proportion of data belonging to a particular meta-domain, the closer the color of the corresponding cell is to blue. We display distribution on some English meta-domains for clarity. The full picture is shown in Figure 7.

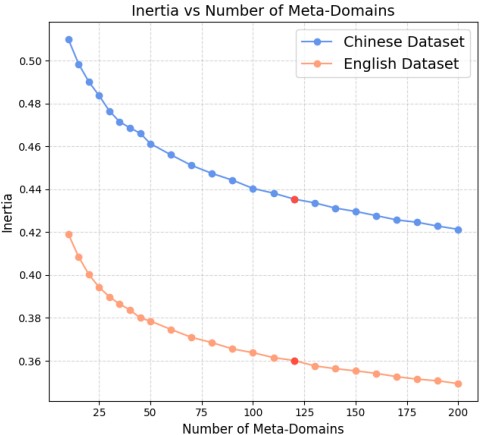

*Figure 2.* The number of meta-domains vs. Inertia.

After getting the corpora, we aim to extract the meta-domains in the corpora, that is, to divide the corpora into some (semantically) different clusters, to serve as the meta-domains. We employ $k$-means (Macqueen, 1967; Arthur & Vassilvitskii, 2006) clustering algorithm to implement the separation and utilize `bge-small-en-v1.5` and `bge-small-zh-v1.5` (Xiao et al., 2023) to compute embeddings for the English and Chinese documents, respectively. See Figure 2 for the relationship between the number of meta-domains and Inertia (measuring the distance between each data point and its centroid). Besides, we divide the code data directly according to the programming language. Ultimately, we construct 260 (120 Chinese + 120 English + 20 Code) unique meta-domains. Each document in the corpora is labeled which meta-domain it originates from.

**Meta-Domain Classifier.** We now present our approach for representing an unseen dataset using the previously established meta-domains. The methodology is straightforward yet effective: we assign each document in the unseen dataset to its corresponding meta-domains and then calculate the aggregate distribution across all documents. This compre-

hensive representation captures the overall domain characteristics of the entire dataset. Formally, assume that there is a meta-domain classifier, for any given document $doc \in \mathcal{D}$,

$$\boldsymbol{p} = [p_1, p_2, p_3, ..., p_n]^\top = \text{Classifier}(doc), \qquad (2)$$

where $p_i$ represents the probability that $doc$ belongs to the $i$-th meta-domain such that $\|\boldsymbol{p}\|_1 = 1$. For the unseen dataset $\mathcal{D}$, we could sample $N$ documents[3] then take the average of domain vector of these samples. Formally, the domain vector $\boldsymbol{v}$ of dataset $\mathcal{D}$ is,

$$\boldsymbol{v} \approx \frac{1}{N} \sum_{i=1}^{N} \boldsymbol{p}_i, \qquad (3)$$

Then, we could use the vector $\boldsymbol{v}$ to approximately represent the feature of any unseen dataset $\mathcal{D}$. Meanwhile, during the pretraining phase of LLMs, we typically have training datasets from many sources $\mathcal{D}_{train} = \{\mathcal{D}_1, \mathcal{D}_2, ..., \mathcal{D}_m\}$. We can convert each of these datasets into domain vectors following Equation 2 and 3. Therefore, $\mathcal{D}_{train}$ can be approximately represented as $\boldsymbol{V}_{train} = [\boldsymbol{v}_1, \boldsymbol{v}_2, ..., \boldsymbol{v}_m]$, where $\boldsymbol{V}_{train} \in \mathbb{R}^{n \times m}$ and $n$ is the number of meta-domains.

Specifically, we train a 260-class classifier to determine which meta-domain a given document originates from. We finetune a `Qwen2-1.5b-base` (Qwen Team, 2024) to balance accuracy and efficiency. After training, the meta-domain classifier achieves a classification accuracy of 74.73% in our constructed test set. For further evaluating the performance of the meta-domain classifier, we also sample 1,000 examples from each sub-dataset of The Pile (Gao et al., 2021). Following Equation 3, we obtain domain vectors predicted by the meta-domain classifier for each sub-dataset, as shown in Figure 1. The distributions of sub-datasets of The Pile over meta-domains exhibit distinctive patterns. This phenomenon indicates not only that

---

[3]In this paper, we set $N = 1000$, which is enough for an accurate and stable domain vector.

the various meta-domains have significant semantic differences, but also that our classifier can accurately distinguish semantic features from different unseen datasets.

# 3. Methodology

In this section, we first introduce the task formulation of optimal data mixture discovery. We then present methodologies for identifying the optimal data mixture using DOMAIN2VEC without requiring additional training. We introduce two approaches: the first is grounded in the *Distribution Alignment Assumption* (**DA$^2$**). Moreover, we demonstrate how our DOMAIN2VEC can be integrated with previous works that model the relationship between mixture ratios and final performance, significantly enhancing the scalability of these existing approaches.

## 3.1. Task Formulation

During the pretraining phase of LLMs, we typically collect training datasets $\mathcal{D}_{train} = \{\mathcal{D}_1, \mathcal{D}_2, ..., \mathcal{D}_m\}$ from $m$ sources (e.g., arXiv, Wikipedia, etc.). We also pre-define a validation set $\mathcal{D}_{valid}$, which is of high quality and corresponding to the final performance. Note that $\mathcal{D}_{valid}$ is often independently and identically distributed with $\mathcal{D}_{train}$. For example, Liu et al. (2024) adopts Pile-CC (Gao et al., 2021) as the validation set and Gu et al. (2024b) adopts LIMA (Zhou et al., 2023) as the validation set. Accordingly, the data mixture $\boldsymbol{r} = [r_1, r_2, ..., r_m]^\top, 0 \leq r_i \leq 1, \sum_{i=1}^{m} r_i = 1$ specifies the mixture ratio of the $m$ datasets. Let the trained LM be denoted as $\theta$, and the validation loss of the LM be denoted as $\mathcal{L}_\theta$. The objective of finding the optimal data mixture $\boldsymbol{r}^*$ is usually to minimize the validation loss, as shown formally in Equation 4. We denote $\mathcal{L}^{\mathcal{D}_{valid}}(\boldsymbol{r})$ as the validation loss of a LM pretrained on the data mixture $\boldsymbol{r}$.

$$\boldsymbol{r}^* = \arg\min_{\boldsymbol{r}}(\min_{\theta} \mathcal{L}_\theta^{\mathcal{D}_{valid}}(\boldsymbol{r})) \triangleq \arg\min_{\boldsymbol{r}} \mathcal{L}^{\mathcal{D}_{valid}}(\boldsymbol{r}) \tag{4}$$

## 3.2. Pilot Study: Mixture Ratio Ranking Holds across Model Sizes

We first conduct a pilot study for a critical research question: *Could the optimal data mixture generalize across different model sizes?* If the answer is *Yes*, it opens up the promising possibility that we could determine the optimal mixture ratio by simply training a small proxy model—or even more efficiently, without training any model at all. To answer the questions, we mix C4 (Raffel et al., 2020) and Knowledge Pile (Fei et al., 2024) with different data mixtures $(0, 0.2, \cdots 1.0)$ in Table 1. We pretrain two LMs with 83M and 1.6B parameters from scratch using the standard LM loss. During pretraining, we evaluate the validation loss of models trained with different mixture ratios on 20 subsets

of The Pile (Gao et al., 2021) and RedPajama (Weber et al., 2024), as shown in Figure 3. The results of more validation sets can be seen in Figures 8 and 9. There are two findings:

- *Given a validation set, there exists an optimal mixture ratio. For different validation sets, the ranking of mixture ratios varies significantly.*

- *For the same validation set, the data mixture ratio ranking does not (nearly) change across model sizes.* We calculate the correlation coefficients of data mixture rankings between the 83M model and the 1.6B model across diverse validation sets. The analysis yields a Spearman coefficient of 0.9743 and a Pearson coefficient of 0.9947, providing robust statistical evidence for this consistency. These exceptionally high correlation values strongly support our finding that optimal mixture ratios are largely invariant to model size when evaluated on the same validation benchmark.

These finding aligns with prior work by Liu et al. (2024), which indicates that it is possible to find the optimal data mixture without training (Section 3.3) or simply training small models (Section 3.4).

## 3.3. Distribution Alignment Assumption (DA$^2$)

We introduce how we directly apply our proposed DOMAIN2VEC on finding optimal data mixture. We notice an intuitive law, that a lower validation loss $\mathcal{L}^{\mathcal{D}_{valid}}$ is achieved when the data distribution of the training set is better aligned with the given validation set[4]. One essential question is that *How do we model the data distribution of various datasets?* Fortunately, according to Section 2, for the training dataset $\mathcal{D}_{train}$, we obtain the vector representation $\boldsymbol{V}_{train} \in \mathbb{R}^{n \times m}$, which models semantic features of $\mathcal{D}_{train}$. Correspondingly, for the validation set $\mathcal{D}_{valid}$, we also have its vector representation $\boldsymbol{v}_{valid}$. After mixing $\mathcal{D}_{train}$ with a data mixture $\boldsymbol{r}$, the final distribution over meta-domains of $\mathcal{D}_{train}$ is $\boldsymbol{v}_{train} = \boldsymbol{V}_{train} \cdot \boldsymbol{r}$. Therefore, based on the distribution alignment assumption, Equation 4 can be equivalently written as:

$$\boldsymbol{r}^* = \arg\min_{\boldsymbol{r}} \text{Dist}(\boldsymbol{V}_{train} \cdot \boldsymbol{r}, \boldsymbol{v}_{valid}) \tag{5}$$

where $\text{Dist}(\cdot, \cdot)$ is a distance function used to measure the similarity between two vectors. Theoretically, numerous distance function options are available, including Wasserstein (optimal transport) distance, Euclidean distance, etc. In this paper, we use Huber Loss (Huber, 1964; Hastie et al., 2009) between two vectors to measure the distance. We also discuss the choice of different distance functions in Appendix E. We present the pseudo code of DOMAIN2VEC+DA$^2$ in Appendix B.

---

[4]We provide the detailed description in the Appendix A.

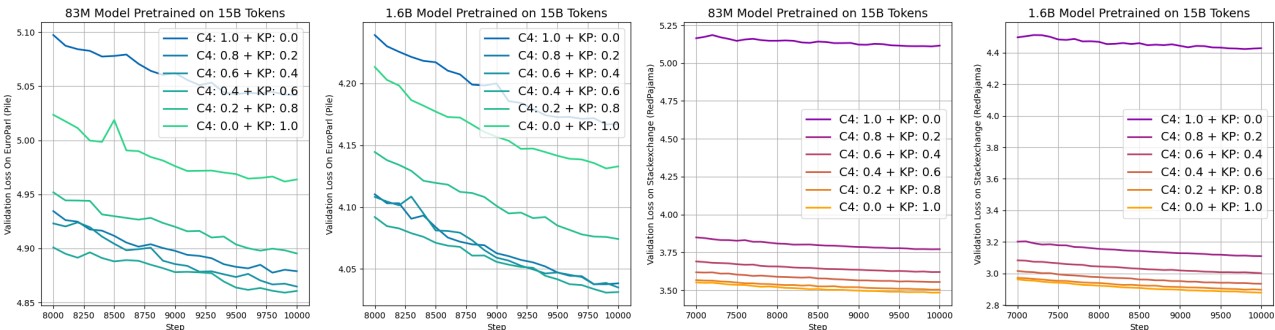

*Figure 3.* The validation loss on the EuroParl (The Pile) and Stackexchange (RedPajama) of models trained using data mixture in Table 1. The validation loss on other validation sets are shown in Appendix D.

### 3.4. Applying Domain2Vec to Prior Work

There is one typical line of research focused on determining optimal mixture ratios, which aims to model the relationship between these ratios and the final validation loss using various functional approaches. That is, these approaches model $\mathcal{L}^{\mathcal{D}_{valid}}(\boldsymbol{r}) = f(\boldsymbol{r})$ where $f(\cdot)$ can take various reasonable forms as proposed in previous works. For example,

- Data Mixing Law (Ye et al., 2024) adopts $f(\boldsymbol{r}) = c_i + k_i \cdot \exp\left(\sum_{j=1}^{m} t_{ij} \cdot r_j\right)$ to predict the validation loss on training domain $i$, where $c_i, k_i, t_{ij}$ are all undetermined parameters to fit.

- RegMix (Liu et al., 2024) initially adopts a Linear Regression approach, modeling the validation loss as $f(\boldsymbol{r}) = \boldsymbol{w}^\top \boldsymbol{r}$ where $\boldsymbol{w}$ needs fitting. Furthermore, it advances this concept by employing LightGBM (Ke et al., 2017) to more effectively fit the function $f(\cdot)$.

We can directly integrate DOMAIN2VEC with these approaches without modifying their core function, but instead perform the computations in the domain vector space. Thereby, we address two inherent limitations of these approaches: (1) **Efficiency**: for modeling $f(\cdot)$ with $m$ variables $r_{1...m}$ [5], it is expected to run experiments $O(m^2)$ times for different $\boldsymbol{r}$ to collect fitting points; (2) **Scalability**: When a new training source is introduced, one must re-collect fitting points and re-fit $f(\cdot)$, which lacks of scalability.

Specifically, we novelly build the relationship $f_i(\cdot)$ between the validation loss on the $i$-th meta-domain $\mathcal{D}_i^*$ (notated as $\mathcal{L}^{D_i^*}$) and the domain vector $\boldsymbol{v}_{train}$ after mixing training datasets by ratio $\boldsymbol{r}$, that is, $\boldsymbol{V}_{train} \cdot \boldsymbol{r}$. Formally, for each meta-domain, we have

$$\mathcal{L}^{\mathcal{D}_i^*}(\boldsymbol{r}) = f_i(\boldsymbol{v}_{train}) = f_i(\boldsymbol{V}_{train} \cdot \boldsymbol{r}), 1 \le i \le n. \quad (6)$$

---

[5] $m$ can scale to to $10^4$ in modern LLM training. For example, Fineweb (Penedo et al., 2024) consists of over 30k data dumps.

Equation 6 enables the prediction of validation loss on any meta-domain given a data mixture, which is also the function that we aim to fit. For unseen validation dataset, recall that any dataset including $\mathcal{D}_{valid}$ can also be viewed as a linear addition of meta-domains and the domain vector of $\mathcal{D}_{valid}$ is denoted as $\boldsymbol{v}_{valid} = [q_1, q_2, \cdots, q_n]^\top$. Therefore, we have

$$
\begin{aligned}
\mathcal{L}^{\mathcal{D}_{valid}}(\boldsymbol{r}) &= \sum_{i=1}^{n} q_i \cdot \mathcal{L}^{\mathcal{D}_i^*}(\boldsymbol{r}) = \sum_{i=1}^{n} q_i \cdot f_i(\boldsymbol{v}_{train}) \\
&= \sum_{i=1}^{n} q_i \cdot f_i(\boldsymbol{V}_{train} \cdot \boldsymbol{r}).
\end{aligned}
\tag{7}
$$

Now, we connect validation loss to the mixture ratio in the the domain vector space via our proposed DOMAIN2VEC. It is feasible to search the optimal mixture ratio $r^\star$ by minimizing $\mathcal{L}^{\mathcal{D}_{valid}}(\boldsymbol{r})$. Note that this connection is built only on the top of the meta-domains (i.e., $f_i$ for $1 \le i \le n$), and can adapt with no cost to (1) any unseen validation set; (2) any unseen training set; (3) any number of training sets. Thanks to this property, we realize the efficiency and scalability by DOMAIN2VEC for prior approaches.

Following RegMix (Liu et al., 2024), we use LightGBM (Ke et al., 2017) as $f(\cdot)$ to fit Equation 6 for each meta-domain (named as DOMAIN2VEC+RegMix). The pseudo code of DOMAIN2VEC + RegMix are shown in Appendix B. We sample $10,500$ diverse mixture ratios from a Dirichlet distribution and we get the validation losses on each meta-domains by training $10,500$ small LMs. We also reserve some mixture ratios as testset and run experiments for evaluating whether fitted function $f(\cdot)$ can accurately predict the validation loss for unseen mixture ratios. For various mixture ratios in the testset, we use the Spearman coefficient to measure the correlation between the predicted ranking and the actual ranking of performance under unseen mixture ratios. Note that we adopt correlation coefficient because it is a more general metric than mean loss error with the goal

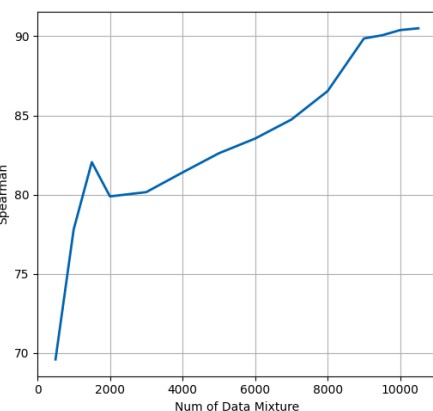

*Figure 4.* Relationship between the number of trained data mixtures and the Spearman correlation.

to find the better mixture ratio than others. Moreover, the pilot study suggests that the predicted ranking holds across model sizes while the predicted loss becomes meaningless for inconsistent model sizes. As shown in Figure 4, the Spearman coefficient increases with the number of mixture ratios that we use for training and collecting fitting points. And finally we get an over 90% Spearman coefficient, which is quite accurate for predicting a good mixture ratio for various meta-domains.

# 4. Experiments

In this section, we elaborate the implementation and experimental results to show how our proposed DOMAIN2VEC helps find the optimal data mixture with less computation. The goal of optimizing the data mixture is to *enhance the performance of LMs*. The performance of LMs can be evaluated from two perspectives: 1) Text generation, often measured by LM loss on a validation set. We aim to minimize the validation loss through finding the optimal data mixture; 2) Downstream task performance. The objective is to optimize task performance. As an overview for exprimental results, by applying DOMAIN2VEC, we can accurately predict the ranking of data mixtures under various settings (e.g., training and validation sets). We also achieve a comparable validation loss with the original data mixture from The Pile but only spend 51.5% training computational resources. Moreover, we use only 0.26% of the computational costs required by DoReMi to find a data mixture with performance comparable to strong baselines like DoReMi.

## 4.1. Validation Loss Minimization

**Dataset & Data Mixture.** We design some training and validation datasets to evaluate the performance to minimize the validation loss of our methods. Our training datasets

include C4 (Raffel et al., 2020) and Knowledge Pile (Fei et al., 2024). C4 is a colossal and cleaned version of Common Crawl corpus. Knowledge Pile is a high-quality dataset that significantly improves the performance of LLMs in knowledge-related and mathematical reasoning tasks. We conduct our experiments on various validation datasets to perform comprehensive evaluation. We select 20 validation datasets from The Pile (Gao et al., 2021) and RedPajama (Weber et al., 2024). Since the optimal mixture ratio varies among the validation datasets, we instead predict the performance ranking across different preset mixture ratios. Specifically, we mix C4 and Knowledge Pile with different data mixtures as the training set as shown in Table 1.

*Table 1.* The preset data mixture ratios.

| Dataset | Data Mixture | | | | | |
|---|---|---|---|---|---|---|
| C4 | 0 | 0.2 | 0.4 | 0.6 | 0.8 | 1.0 |
| Knowledge Pile | 1.0 | 0.8 | 0.6 | 0.4 | 0.2 | 0.0 |

**Training & Evaluation Setup.** We pretrain LLaMA-like (Grattafiori et al., 2024) models with 83M and 1.6B parameters from scratch using standard language modeling loss. Both models have a batch size of 1.5M tokens and a maximum sequence length of 4,096. We use the AdamW optimizer (Loshchilov & Hutter, 2017) with gradient clipping at 1.0. The learning rate linearly warms up to 2e-4 over the first 100 steps, then decays to 2e-5 using a cosine scheduler over 10,000 steps. More parameters are detailed in Table 7. Then, we evaluate DOMAIN2VEC using the Spearman and Pearson correlation coefficient between the predicted ranking and the actual ranking. We compare DOMAIN2VEC with randomly ranking and an embedding-based baseline, denoted as $k$NN. Specifically, we use `bge-small-v1.5` to obtain embeddings and apply mean pooling to generate unique embeddings for each dataset and meta-domain. We then apply $k$NN based on Euclidean distance to compute the probability distributions of training and test datasets originating from each meta-domain, treating these distributions as new domain vectors.

**Experimental Results.** We present the validation loss curves for various data mixtures in Figure 3 and Appendix D. It can be observed that, on most validation sets, incorporating a certain amount of Knowledge Pile significantly reduces the validation loss, even on the C4 validation set from RedPajama. We apply two DOMAIN2VEC-based methods described in Section 3 to rank the data mixture from Table 1.

As demonstrated in Table 2, the ranking predicted by DOMAIN2VEC exhibits a strong positive correlation with the actual ranking, significantly outperforming random guessing and $k$NN. The effectiveness of the $k$NN method partially validates the rationale behind our meta-domain vocabulary construction. It is also important to note that our method

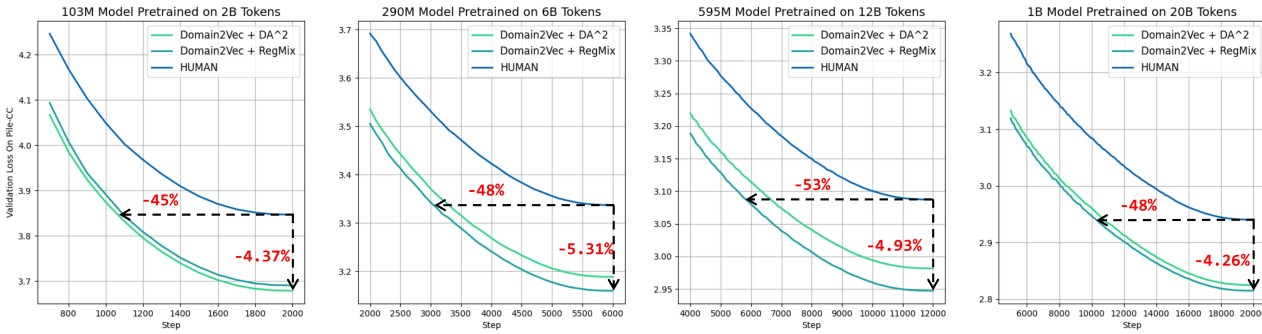

*Figure 5.* The validation loss on the Pile-CC subset. DOMAIN2VEC achieves the comparable validation loss of Human (The model using original data mixture from The Pile), which only uses almost 51.5% training computational costs of Human. Using the same training cost, DOMAIN2VEC can reduce the validation loss by approximately 4.72% compared to Human.

is a *training-free approach*, unlike prior works that rely on training small proxy models to rank data mixtures. Despite the more challenging setup, our method accurately predicts the rankings of different data mixtures.

*Table 2.* The results of deploying the DOMAIN2VEC to predict the ranking of different Validation sets.

| Metrics | Random | *k*NN | DOMAIN2VEC+DA$^2$ | DOMAIN2VEC+RegMix |
|---|---|---|---|---|
| Pearson | 0.0300 | 0.4014 | **0.5833** | 0.3881 |
| Spearman | 0.0497 | 0.3543 | **0.6657** | 0.4629 |

### 4.2. Downstream Task Performance Maximization

In this section, we demonstrate how DOMAIN2VEC can be used to identify the optimal data mixture for maximizing downstream task performance. One challenge is modeling the relationship between data mixture and downstream performance. Fortunately, Liu et al. (2024) finds that *validation loss on Pile-CC correlates most strongly with downstream performance across their evaluations*. To align with prior work, we follow and use the same validation datasets as Liu et al. (2024). Thus, our goal is to identify a data mixture that minimizes validation loss on Pile-CC. *Experimental results show that DOMAIN2VEC predicts a data mixture with performance comparable to DoReMi (Xie et al., 2023a), while using only 0.26% computational cost.*

**Datasets & Baselines.** We follow RegMix (Liu et al., 2024) and use The Pile (Gao et al., 2021) as our training datasets. The Pile is an 825 GB English text corpus used for LLM pretraining. In line with RegMix, we use only the 17 components of The Pile that do not have copyright issues. Our goal is to identify the data mixture that minimizes validation loss on the Pile-CC subset to improve downstream task performance. We compare our approach with several baselines, including Human (the original data mixture), DoReMi (Xie et al., 2023a), and RegMix (Liu

et al., 2024). The Pile-CC Only baseline (which trains the model solely on the Pile-CC subset) is included to verify the strong correlation between Pile-CC validation loss and downstream performance. The data mixtures for each baseline are shown in Table 5.

**Training & Evaluation Setup.** We pretrain LLaMA-like (Grattafiori et al., 2024) models from scratch using standard language modeling loss with model sizes ranging from 106M to 1B parameters. Following Hoffmann et al. (2022a), the token count for each model is 20 times corresponding parameter size. All models adopt a batch size of 1M tokens and a maximum sequence length of 4,096. We apply AdamW (Loshchilov & Hutter, 2017) optimizer with gradient clipping at 1.0. The learning rate linearly warms up to 6e-4 over 1,000 steps, then decays to 0 using a cosine scheduler at the end of training. More parameters are detailed in Table 7. For evaluation, we track the performance on Pile-CC validation loss across different model sizes. Besides, we evaluate the performance of different data mixture using following benchmarks: Social IQA (Sap et al., 2019), HellaSwag (Zellers et al., 2019), PiQA (Bisk et al., 2019), OpenBookQA (Mihaylov et al., 2018), Lambada (Paperno et al., 2016), SciQ (Welbl et al., 2017), ARC Easy (Clark et al., 2018), COPA (Gordon et al., 2012), RACE (Lai et al., 2017), LogiQA (Liu et al., 2021), WinoGrande (Sakaguchi et al., 2021), and MultiRC (Khashabi et al., 2018). We utilize LM Evaluation Harness (Gao et al., 2024) to evaluate these models and report the average score across 0-shot to 5-shot settings in Table 3.

**Implementation Details.** We predict the optimal data mixture by applying Equation 5 (DOMAIN2VEC+DA$^2$) and Equation 7 (DOMAIN2VEC+RegMix). We generate 100,000 data mixtures from a Dirichlet distribution based on the token distribution of these components. Using these mixtures, we predict the optimal data mixture by our proposed two methods. We select top-100 predicted data mix-

*Table 3.* **Average** downstream task performance of different models pretrained on different data mixtures. Similar to Liu et al. (2024), Human refers the original data mixture from The Pile. Pile-CC is a golden training set which can 100% correspond to validation set to validate our propose $DA^2$. All the data mixtures are shown in Table 5 and Table 6. The calculated data mixture are shown in Table 8.

| Benchmark | Human | DoReMi | Pile-CC Only | RegMix | DOMAIN2VEC + DA$^2$ | DOMAIN2VEC + RegMix |
|---|---|---|---|---|---|---|
| Social IQA | 0.367 | 0.380 | 0.381 | 0.382 | 0.372 | 0.375 |
| HellaSwag | 0.319 | 0.346 | 0.351 | 0.351 | 0.335 | 0.338 |
| PiQA | 0.615 | 0.639 | 0.644 | 0.647 | 0.635 | 0.639 |
| OpenBookQA | 0.264 | 0.275 | 0.276 | 0.276 | 0.275 | 0.272 |
| Lambada | 0.199 | 0.240 | 0.247 | 0.241 | 0.219 | 0.232 |
| SciQ | 0.710 | 0.695 | 0.688 | 0.708 | 0.701 | 0.701 |
| ARC Easy | 0.411 | 0.428 | 0.436 | 0.438 | 0.427 | 0.426 |
| COPA | 0.621 | 0.651 | 0.660 | 0.653 | 0.638 | 0.641 |
| RACE | 0.274 | 0.291 | 0.288 | 0.288 | 0.279 | 0.282 |
| LogiQA | 0.272 | 0.275 | 0.272 | 0.272 | 0.269 | 0.278 |
| WinoGrande | 0.512 | 0.516 | 0.515 | 0.513 | 0.513 | 0.510 |
| MultiRC | 0.521 | 0.528 | 0.515 | 0.529 | 0.524 | 0.534 |
| Average Performance | 0.424 | 0.439 | 0.439 | 0.441 | 0.432 | 0.436 |
| Estimated FLOPs | 0 | $3.7 \times 10^{19}$ (100%) | 0 | $3.5 \times 10^{18}$ (9.46%) | $9.66 \times 10^{16}$ (0.26%) | $9.66 \times 10^{16}$ (0.26%) |

tures and average them as the final data mixture. This trick is aligned with previous work (Liu et al., 2024) for more accurate and stable results. As a stardard practice, each subset of The Pile is trained for at most one epoch. When optimizing the mixture ratio $r = [r_1, r_2, \cdots, c_m]^\top$, other than the trivial restriction $\sum_{i=1}^{m} r_i = 1$, note that there is another data amount restriction, that is $\#TotalTokens \cdot r_i \leq |\mathcal{D}_i|$, which is to remove data mixtures which require exceeding tokens in some subsets. Therefore the optimal data mixture predicted by DOMAIN2VEC may vary depending on the number of trained tokens, as well as the size of the model. This restriction is different with Section 4.1 where each dataset size is seen as unlimited.

**Experimental Results.** As shown in Figure 5, *our proposed* DOMAIN2VEC + DA$^2$ *and* DOMAIN2VEC + REG-MIX *significantly improve training efficiency on Pile-CC compared to Human*. Specifically, DOMAIN2VEC + DA$^2$ and DOMAIN2VEC + REGMIX require only about 55.38% and 51.50% of the training steps, respectively, to achieve the same validation loss as Human. Compared to Human under the same compute budget, DOMAIN2VEC + DA$^2$ and DOMAIN2VEC + REGMIX reduce validation loss by approximately 4.04% and 4.64%, and improves downstream performance by an average of 1.89% and 2.83%, respectively. In Table 3, we report the average performance of LMs trained on data mixtures from various baselines across a range of downstream tasks. "Pile-CC only" shows a 3.54% average accuracy improvement over Human, indicating that training on more tokens from Pile-CC enhances downstream performance. Importantly, "Pile-CC only" is good when we regard Pile-CC as validation set. However, in a more practical scenario where validation set is somewhat else, we cannot manually find such a golden training set which

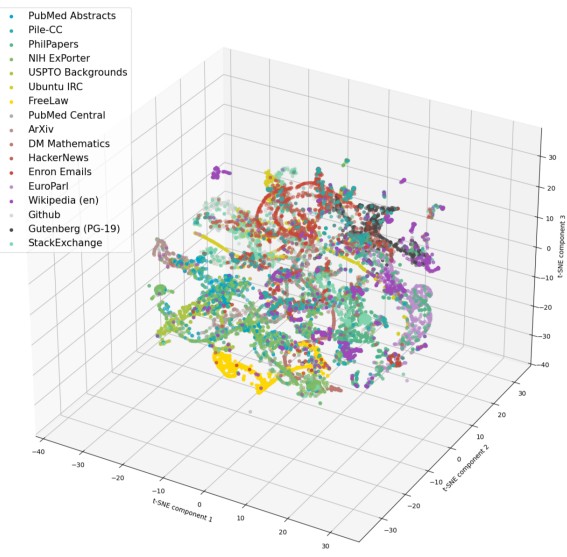

*Figure 6.* Visualization (t-SNE) of domain vectors of The Pile.

can 100% correspond to validation set. To this end, we can use our proposed Domain2Vec to get a comparable downstream performance with lowest cost by mixing datasets from different sources. Notably, DOMAIN2VEC + DA$^2$ *and* DOMAIN2VEC + REGMIX, *using only about 0.26% of the FLOPs required by DoReMi, achieve performance comparable to DoReMi, RegMix*, which demonstrates the computational efficiency of DOMAIN2VEC.

**Visualization.** To investigate further, we employ t-SNE (Van der Maaten & Hinton, 2008) to visualize the domain vectors of each component in The Pile, as shown in Figure 6. This visualization reveals several desirable properties of the learned vectors. The representation space

exhibits strong clustering behavior where semantically related datasets naturally group together, indicating effective capture of domain-specific characteristics. Related domains such as academic literature (PubMed, arXiv) and technical repositories (GitHub, StackExchange) demonstrate spatial coherence, while maintaining well-defined yet flexible boundaries between different domains. The representation spans diverse domains in The Pile, demonstrating robust generalization capabilities across heterogeneous data types.

### 4.3. Discussion on Overfitting

We noticed that some readers interpret our approach $DA^2$ as a form of "overfitting": optimizing on a selected validation set. We offer the following explanations:

- The validation set that we define is actually a guide dataset, which is a necessary requirement for optimization data mixture and a common setting in related works (see Section 3.1).

- In Section 4.1, we conduct experiments on various validation sets, and the performance demonstrates good stability. In fact, our proposed $DA^2$ does not even require training, thus "overfitting" is not applicable.

- In Section 4.2, we choose Pile-CC as the validation set but ultimately test model performance on benchmarks from 12 downstream tasks, further preventing overfitting risks.

## 5. Related Work

Recent research on optimizing data mixture can be broadly divided into two lines. The first line implicitly adjusts data mixture by down-sampling from various datasets based on data quality. For example, Lin et al. (2024) propose RHO-1, which uses Selective Language Models to select tokens that align the data mixture with the ideal ratio. Instead of token-level selection, Ankner et al. (2024) filter low-quality samples using the perplexity of small reference models. Thakkar et al. (2023) demonstrate that the Influence Score can guide data re-weighting, while their subsequent work introduces an online data selection method that eliminates the need for reference models.

The second line focuses on explicitly adjusting data mixture by modeling the relationship between data mixture and language model performance. The simplest approach is to observe the performance of various data mixtures and select the optimal one, as done during Gopher training (Rae et al., 2022). This is costly and difficult to scale for larger models. Xie et al. (2023a) propose DoReMi and use a small proxy model to re-weight data from different domains, improving training efficiency for larger models. However, DoReMi still requires a pre-trained reference model, adding

computational costs and making it hard to define an ideal reference model. Some works aim to model the functional relationship between data mixture and the LM performance. Inspired by scaling laws (Kaplan et al., 2020; Hoffmann et al., 2022a), Ye et al. (2024) introduce Data Mixing Laws, which describe this relationship using an exponential form. Ge et al. (2024) propose BiMix, a scaling law that considers both compute and data mixture. Que et al. (2024) and Wang et al. (2025) develop scaling laws for continual pretraining, and how mixture ratio as one variable impacts LM loss is modeled. Recently, Liu et al. (2024) propose Linear Regression to model the validation loss across different data mixtures, showing a strong and promising performance.

All these prior works face two main issues: **1) Computational Efficiency**: For example, the estimated FLOPs for DoReMi and RegMix are high to $3.7 \times 10^{19}$ and $3.5 \times 10^{18}$, respectively, for calculating less than 22 datasets. Moreover, the computational complexity of these methods will grow non-linearly as the number of datasets increases. **2) Lack of Scalability**: When the components of the training dataset change (e.g., adding some new datasets), previous methods (Ye et al., 2024; Liu et al., 2024) require resampling data mixtures, retraining proxy models, and then re-performing the fitting process. In this paper, we introduce DOMAIN2VEC to decompose any dataset into a linear combination of meta-domains. DOMAIN2VEC shares some concepts with prior meta-learning works, such as Jomaa et al. (2021) and Chen et al. (2024), which explore dataset representation in latent spaces. While sharing this concept, DOMAIN2VEC differs in both purpose and implementation, and we focus on the data mixture in LM pretraining.

## 6. Conclusion

In this work, we introduce DOMAIN2VEC, a novel method to capture the underlying features of datasets by decomposing datasets into a linear combination of several meta-domains. It enables us to acquire vectorized representation for arbitrary datasets. Building on these domain vectors, we introduce a training-free approach by Distribution Alignment Assumption ($DA^2$) to identify optimal data mixtures for language model pretraining Furthermore, DOMAIN2VEC seamlessly integrates with existing methods, significantly enhancing their efficiency and scalability while establishing a direct relationship between model performance and computed domain vectors–all without requiring retraining when training datasets change. Our experimental results demonstrate that both DOMAIN2VEC+$DA^2$ and DOMAIN2VEC+RegMix achieve comparable text generation and downstream task performance with reduced computational overhead compared to existing approaches. We believe this work offers valuable insights into optimizing data mixtures for language model pretraining and paves the way for more efficient training strategies.

## Impact Statement

This paper presents work whose goal is to advance the field of Machine Learning. There are many potential societal consequences of our work, none which we feel must be specifically highlighted here.

## Acknowledgements

This work was supported by the National Natural Science Foundation of China (No. U24B20181) and Fujian Provincial Natural Science Foundation of China (No. 2024J08371).

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

## A. Detailed Description of the Distribution Alignment Assumption

In this section, we will introduce the detailed description of the Distribution Alignment Assumption for language model pretraining.

In the scenario of finding the optimal data mixture for language model pretraining, the validation set $\mathcal{D}_{valid}$ is fixed, and we should adjust the data mixture to construct the training set $\mathcal{D}_{train}$ to achieve lower validation loss calculated by Equation 8, where $\hat{\theta}$ is parameters of a pretrained language model.

$$\mathbb{E}_{X \sim \mathcal{D}_{valid}} - \log P(X|\hat{\theta}) = \mathbb{E}_{X \sim \mathcal{D}_{valid}} \sum_{i=1}^{|X|} - \log(P(x_i|x_{<i}, \hat{\theta})) \tag{8}$$

Typically, we pretrain language models via next token prediction (Radford, 2018) like Equation 9.

$$\hat{\theta} = \arg\max_{\theta} \mathbb{E}_{X \sim \mathcal{D}_{train}} \log P(X|\theta)$$

$$= \arg\max_{\theta} \mathbb{E}_{X \sim \mathcal{D}_{train}} \sum_{i=1}^{|X|} \log(P(x_i|x_{<i}, \theta)) \tag{9}$$

That is, we need to find a $\hat{\theta}$ that maximizes the expected probability of $X \sim \mathcal{D}_{train}$, which is also known as Maximum Likelihood Estimation (MLE). When the data distributions of $\mathcal{D}_{\text{train}}$ and $\mathcal{D}_{\text{valid}}$ are aligned, the optimization target of language models pretraining process equals find a $\hat{\theta}$ that maximizes the expected probability of $X \sim \mathcal{D}_{valid}$. Therefore, we introduce the Distribution Alignment Assumption for language model pretraining, a novel method to find the optimal data mixture without training.

## B. Algorithm

In Algorithm 1, we show the pseudo code for acquiring the domain vector for pretraining datasets.

In Algorithm 2 and 3, we show the pseudo code for how to use DOMAIN2VEC to find the optimal data mixture, including Distribution Alignment Assumption, and applying DOMAIN2VEC to RegMix.

Note that when applying DOMAIN2VEC+DA$^2$ or DOMAIN2VEC+RegMix , for getting more stable and accurate results, one could also average the $k$-best ratios in the K sampled candidates data mixture. We present top-1 as one example in the pseudo codes. We adopt top-1 for direct comparison in Section 4.1, while we adopt top-100 in Section 4.2, which is aligned with RegMix (Liu et al., 2024).

---

**Algorithm 1** DOMAIN2VEC

**Require:** Training datasets $\mathcal{D}_{train} = \{\mathcal{D}_1, \mathcal{D}_2, ..., \mathcal{D}_m\}$ , validation dataset $\mathcal{D}_{valid}$, meta-domain classifier Classifier
 1: Domain vectors $V_{train} = []$
 2: **for** $i = 1$ **to** $m$ **do**
 3:     Sample $N$ documents $doc_{1...N}$ from $\mathcal{D}_i$
 4:     $\boldsymbol{v}_i = \frac{1}{N} \sum_{j=1}^N \text{Classifier}(doc_j)$, where $doc_j \in \mathcal{D}_i$
 5:     $\boldsymbol{V}_{train} = [\boldsymbol{V}_{train}, \boldsymbol{v}_i]$
 6: **end for**
 7: Sample $N$ data points from $\mathcal{D}_{valid}$
 8: $\boldsymbol{v}_{valid} = \frac{1}{N} \sum_{j=1}^N \text{Classifier}(doc_j)$, where $doc_j \in \mathcal{D}_{valid}$
 9: **Return:** $\boldsymbol{V}_{train} = [v_1, v_2, ..., v_m], \boldsymbol{v}_{valid}$

---

## C. Data Mixture of Different Methods

In this section, we will show the data mixture on The Pile (Gao et al., 2021) of different methods we used in this paper for reproduction. In Table 5, we show the optimal data mixture predicted by DOMAIN2VEC + DA$^2$ and DOMAIN2VEC + RegMix. It should be noted that, to avoid the over-fitting problem, any subset of The Pile (Gao et al., 2021) will be only trained at most one epoch. Because we adopt rejection sampling to filter out certain unreasonable data mixtures. The data mixture predicted may change as model sizes change.

---

**Algorithm 2** DOMAIN2VEC+DA$^2$

---

**Require:** Domain vectors of training datasets $\boldsymbol{V}_{train} = [\boldsymbol{v}_1, \boldsymbol{v}_2, ..., \boldsymbol{v}_m]$, domain vectors of validation dataset $\boldsymbol{v}_{valid}$, token distribution of training datasets $\boldsymbol{a}_{train}$.

1: Sample $K$ candidates data mixture $\boldsymbol{r}_i$ from Dirichlet($\boldsymbol{a}_{train}$)
2: The optimal data mixture $\boldsymbol{r}^* = \boldsymbol{r}_1$
3: **for** $i = 2$ **to** $K$ **do**
4:    **if** $\text{Dist}(\boldsymbol{V}_{train} \cdot \boldsymbol{r}, \boldsymbol{v}_{valid}) < \text{Dist}(\boldsymbol{V}_{train} \cdot \boldsymbol{r}^*, \boldsymbol{v}_{valid})$ **then**
5:       $\boldsymbol{r}^* = \boldsymbol{r}_i$
6:    **end if**
7: **end for**
8: **Return:** the optimal data mixture $\boldsymbol{r}^*$

---

---

**Algorithm 3** DOMAIN2VEC+RegMix

---

**Require:** Domain vectors of training datasets $\boldsymbol{V}_{train} = [\boldsymbol{v}_1, \boldsymbol{v}_2, \cdots, \boldsymbol{v}_m]$, domain vectors of validation dataset $\boldsymbol{v}_{valid} = [q_1, q_2, \cdots, q_n]^\top$, token distribution of training datasets $\boldsymbol{a}_{train}$, fitted model for each meta-domain $f_i(\cdot)$.

1: Sample $K$ candidates data mixture $r_i$ from Dirichlet($\boldsymbol{a}_{train}$)
2: The optimal data mixture $r^* = r_1$
3: Def $\mathcal{L}(\boldsymbol{r}) = \sum\limits_{i=1}^{n} q_i \cdot f_i(\boldsymbol{V}_{train} \cdot \boldsymbol{r})$
4: **for** $i = 2$ **to** $K$ **do**
5:    **if** $\mathcal{L}(\boldsymbol{r}_i) < \mathcal{L}(\boldsymbol{r}^*)$ **then**
6:       $\boldsymbol{r}^* = \boldsymbol{r}_i$
7:       $\mathcal{L}(\boldsymbol{r}^*) = \mathcal{L}(\boldsymbol{r}_i)$
8:    **end if**
9: **end for**
10: **Return:** the optimal data mixture $\boldsymbol{r}^*$

---

## D. Experimental Results of Pilot Study

In this section, we report the validation loss on various datasets arXiv, C4, Book3, PG19 from RedPajama (Weber et al., 2024), and BookCorpus2, DM Mathematics, Enron Emails, FreeLaw, HackerNews, NIH ExPorter, OpenSubtitles, OpenWebText2, PhilPapers, PubMed Abstracts, PubMed Central, USPTO Backgrounds, Ubuntu IRC, Youtube Subtitles from The Pile (Gao et al., 2021) in Figure 3, Figure 8 and Figure 9.

## E. Comparative Study on Different Distributional Measures of DA$^2$

In Section 3.3, we use Huber Loss to measure the similarity of domain vectors. Technically, Huber loss combines the advantages of L1 and L2 distance. In Table 4, we add the results of different distributional measures. As shown in the Table 4, Huber Loss shows better performance than L1/L2/JS Distance. Additionally, Wasserstein distance is a very great option. However, it would require an extra metric space matrix, $\boldsymbol{M}$, to measure the distance between two domain vectors. In this work, the metric space, $\boldsymbol{M} \in \mathbb{R}^{260 \times 260}$, is actually the "dataset transition cost" between each two meta-domains, and is non-trivial. Each element in $\boldsymbol{M}$. $c_{i,j}$ could be estimated via $\mathcal{L}_{i,j}$, the loss at meta-domain $j$ after training on meta-domain $i$, which requires additional computational resources. Considering that Huber Loss already achieved very positive results, we did not conduct this experiment. We believe that Wasserstein distance can also present a positive result (even better) if the metric space is well estimated, and we leave this for future work.

*Table 4.* Huber Loss shows better performance than L1/L2/JS Distance.

| Distributional Measure | Pearson | Spearman |
|---|---|---|
| Huber Loss | 0.5833 | 0.6657 |
| JS Distance | 0.4527 | 0.5000 |
| L1 Distance | 0.4830 | 0.5400 |
| L2 Distance | 0.5720 | 0.6429 |

*Table 5.* The data mixture of The Pile (Gao et al., 2021) from different baselines, which aligns with the data mixture used in Liu et al. (2024).

| Data Mixture | Human | DoReMi | Pile-CC Only | RegMix |
|---|---|---|---|---|
| ArXiv | 0.134 | 0.004 | 0.0 | 0.001 |
| FreeLaw | 0.049 | 0.005 | 0.0 | 0.001 |
| NIH ExPorter | 0.007 | 0.008 | 0.0 | 0.001 |
| PubMed Central | 0.136 | 0.006 | 0.0 | 0.003 |
| Wikipedia (en) | 0.117 | 0.086 | 0.0 | 0.016 |
| DM Mathematics | 0.025 | 0.002 | 0.0 | 0.0 |
| Github | 0.054 | 0.022 | 0.0 | 0.0 |
| PhilPapers | 0.003 | 0.034 | 0.0 | 0.0 |
| Stack Exchange | 0.118 | 0.019 | 0.0 | 0.0 |
| Enron Emails | 0.004 | 0.009 | 0.0 | 0.002 |
| Gutenberg (PG-19) | 0.025 | 0.009 | 0.0 | 0.002 |
| Pile-CC | 0.142 | 0.743 | 1.0 | 0.87 |
| Ubuntu IRC | 0.009 | 0.011 | 0.0 | 0.064 |
| EuroParl | 0.005 | 0.008 | 0.0 | 0.0 |
| HackerNews | 0.01 | 0.016 | 0.0 | 0.012 |
| PubMed Abstracts | 0.107 | 0.014 | 0.0 | 0.024 |
| USPTO Backgrounds | 0.053 | 0.004 | 0.0 | 0.002 |

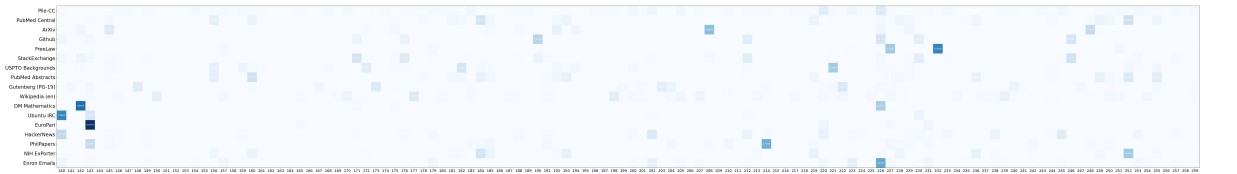

*Figure 7.* The Domain Vector of each sub-dataset of The Pile (Gao et al., 2021), where each row corresponds to a sub-dataset and each column corresponds to a meta-domain. The higher the proportion of data belonging to a particular meta-domain, the closer the color of the corresponding cell is to blue). Additionally, since The Pile primarily consists of English texts, we only display the distribution on English meta-domains for clarity.

*Table 6.* The optimal data mixture predicted by DOMAIN2VEC + DA$^2$ and DOMAIN2VEC + RegMix. To avoid the over-fitting problem, any subset of The Pile (Gao et al., 2021) will be trained at most one epoch. And we adopt rejection sampling to filter out certain unreasonable data mixtures. Thus, the data mixture predicted may change as model sizes change.

| Data Mixture | DOMAIN2VEC+DA$^2$ | | | | DOMAIN2VEC+RegMix | | | |
|---|---|---|---|---|---|---|---|---|
| | 106M | 290M | 595M | 1B | 106M | 290M | 595M | 1B |
| ArXiv | 0.0131 | 0.0131 | 0.0389 | 0.0431 | 0.0152 | 0.0070 | 0.0114 | 0.0103 |
| FreeLaw | 0.0076 | 0.0076 | 0.0316 | 0.0305 | 0.0395 | 0.0267 | 0.0339 | 0.0268 |
| NIH ExPorter | 0.0008 | 0.0008 | 0.0028 | 0.0023 | 0.0000 | 0.0199 | 0.0000 | 0.0000 |
| PubMed Central | 0.0773 | 0.0773 | 0.0519 | 0.0704 | 0.0343 | 0.0576 | 0.0099 | 0.0518 |
| Wikipedia (en) | 0.2970 | 0.2970 | 0.2049 | 0.2126 | 0.0847 | 0.0101 | 0.1014 | 0.2577 |
| DM Mathematics | 0.0003 | 0.0003 | 0.0056 | 0.0026 | 0.0177 | 0.0018 | 0.0011 | 0.0008 |
| Github | 0.0096 | 0.0096 | 0.0290 | 0.0298 | 0.0034 | 0.0538 | 0.0500 | 0.0138 |
| PhilPapers | 0.0018 | 0.0018 | 0.0093 | 0.0025 | 0.0118 | 0.0005 | 0.0333 | 0.0401 |
| Stack Exchange | 0.0464 | 0.0464 | 0.0661 | 0.0585 | 0.0698 | 0.0430 | 0.1199 | 0.0262 |
| Enron Emails | 0.0000 | 0.0000 | 0.0009 | 0.0000 | 0.0018 | 0.0000 | 0.0000 | 0.0000 |
| Gutenberg (PG-19) | 0.0217 | 0.0217 | 0.0484 | 0.0370 | 0.0467 | 0.0223 | 0.0007 | 0.0252 |
| Pile-CC | 0.4338 | 0.4338 | 0.3191 | 0.3814 | 0.5370 | 0.6323 | 0.5546 | 0.4704 |
| Ubuntu IRC | 0.0022 | 0.0022 | 0.0063 | 0.0072 | 0.1019 | 0.0123 | 0.0161 | 0.0069 |
| EuroParl | 0.0003 | 0.0003 | 0.0042 | 0.0040 | 0.0070 | 0.0037 | 0.0116 | 0.0000 |
| HackerNews | 0.0154 | 0.0154 | 0.0521 | 0.0199 | 0.0028 | 0.0551 | 0.0170 | 0.0673 |
| PubMed Abstracts | 0.0596 | 0.0596 | 0.0739 | 0.0532 | 0.0259 | 0.0102 | 0.0190 | 0.0017 |
| USPTO Backgrounds | 0.0130 | 0.0130 | 0.0549 | 0.0449 | 0.0004 | 0.0438 | 0.0201 | 0.0010 |

*Table 7.* The parameters of different models we used in Section 4.1 and Section 4.2. When calculating the model parameters, we do not take into account the embedding layer and the language model head layer.

| Parameter | Text Generation | | Downstream Task | | | |
|---|---|---|---|---|---|---|
| | 83M | 1.6B | 106M | 290M | 595M | 1B |
| Hidden Size | 768 | 2,048 | 768 | 1,280 | 1,536 | 2,048 |
| FFN Hidden Size | 2,048 | 5,504 | 2,048 | 3,392 | 4,096 | 5,440 |
| Num of Layers | 12 | 24 | 15 | 15 | 21 | 21 |
| Num of Heads | 12 | 16 | 12 | 10 | 12 | 32 |
| Max Seq Length | 4,096 | 4,096 | 4,096 | 4,096 | 4,096 | 4,096 |
| Vocab Size | 128,256 | 128,256 | 151,936 | 151,936 | 151,936 | 151,936 |
| RoPE Base | 10,000 | 10,000 | 10,000 | 10,000 | 10,000 | 10,000 |

*Table 8.* Downstream Task Performance of different data mixture on 106M Model. Similar to Liu et al. (2024), Human refers the original data mixture from The Pile. Pile-CC Only refers only training on the Pile-CC subset. The data mixture and estimated flops of DoReMi and RegMix are from Liu et al. (2024).

| Benchmark | Human | DoReMi | Pile-CC Only | RegMix | DOMAIN2VEC + DA$^2$ | DOMAIN2VEC + RegMix |
|---|---|---|---|---|---|---|
| *106M Model Pretrained on 2B Tokens* | | | | | | |
| Social IQA | 0.340 | 0.349 | 0.353 | 0.356 | 0.339 | 0.342 |
| HellaSwag | 0.268 | 0.268 | 0.269 | 0.269 | 0.267 | 0.264 |
| PiQA | 0.573 | 0.584 | 0.580 | 0.586 | 0.579 | 0.583 |
| OpenBookQA | 0.245 | 0.251 | 0.249 | 0.242 | 0.245 | 0.249 |
| Lambada | 0.065 | 0.099 | 0.102 | 0.091 | 0.091 | 0.090 |
| SciQ | 0.550 | 0.520 | 0.509 | 0.537 | 0.549 | 0.518 |
| ARC Easy | 0.329 | 0.339 | 0.335 | 0.337 | 0.334 | 0.331 |
| COPA | 0.525 | 0.570 | 0.572 | 0.585 | 0.578 | 0.557 |
| RACE | 0.236 | 0.254 | 0.246 | 0.251 | 0.240 | 0.244 |
| LogiQA | 0.282 | 0.280 | 0.271 | 0.274 | 0.268 | 0.286 |
| WinoGrande | 0.516 | 0.516 | 0.502 | 0.508 | 0.506 | 0.499 |
| MultiRC | 0.539 | 0.520 | 0.515 | 0.533 | 0.541 | 0.544 |
| Average Performance | 0.372 | 0.379 | 0.375 | 0.381 | 0.378 | 0.376 |
| *290M Model Pretrained on 6B Tokens* | | | | | | |
| Social IQA | 0.364 | 0.373 | 0.374 | 0.371 | 0.371 | 0.368 |
| HellaSwag | 0.295 | 0.312 | 0.317 | 0.315 | 0.307 | 0.312 |
| PiQA | 0.605 | 0.631 | 0.639 | 0.642 | 0.624 | 0.633 |
| OpenBookQA | 0.261 | 0.271 | 0.271 | 0.262 | 0.268 | 0.266 |
| Lambada | 0.175 | 0.208 | 0.206 | 0.210 | 0.182 | 0.208 |
| SciQ | 0.711 | 0.682 | 0.663 | 0.674 | 0.670 | 0.697 |
| ARC Easy | 0.395 | 0.410 | 0.419 | 0.417 | 0.420 | 0.412 |
| COPA | 0.632 | 0.660 | 0.682 | 0.657 | 0.627 | 0.642 |
| RACE | 0.265 | 0.280 | 0.280 | 0.276 | 0.283 | 0.281 |
| LogiQA | 0.283 | 0.293 | 0.296 | 0.276 | 0.277 | 0.292 |
| WinoGrande | 0.511 | 0.506 | 0.509 | 0.524 | 0.498 | 0.504 |
| MultiRC | 0.507 | 0.555 | 0.513 | 0.545 | 0.521 | 0.517 |
| Average Performance | 0.417 | 0.432 | 0.431 | 0.431 | 0.421 | 0.428 |
| *595M Model Pretrained on 12B Tokens* | | | | | | |
| Social IQA | 0.378 | 0.387 | 0.390 | 0.394 | 0.383 | 0.388 |
| HellaSwag | 0.338 | 0.377 | 0.386 | 0.385 | 0.355 | 0.366 |
| PiQA | 0.624 | 0.656 | 0.663 | 0.667 | 0.651 | 0.659 |
| OpenBookQA | 0.273 | 0.279 | 0.283 | 0.294 | 0.288 | 0.271 |
| Lambada | 0.255 | 0.294 | 0.332 | 0.310 | 0.269 | 0.292 |
| SciQ | 0.777 | 0.757 | 0.770 | 0.791 | 0.763 | 0.769 |
| ARC Easy | 0.439 | 0.453 | 0.478 | 0.481 | 0.453 | 0.460 |
| COPA | 0.642 | 0.680 | 0.672 | 0.663 | 0.668 | 0.667 |
| RACE | 0.289 | 0.309 | 0.311 | 0.311 | 0.288 | 0.303 |
| LogiQA | 0.263 | 0.268 | 0.252 | 0.267 | 0.263 | 0.267 |
| WinoGrande | 0.509 | 0.515 | 0.506 | 0.509 | 0.512 | 0.503 |
| MultiRC | 0.516 | 0.533 | 0.522 | 0.507 | 0.506 | 0.527 |
| Average Performance | 0.442 | 0.459 | 0.464 | 0.465 | 0.450 | 0.456 |
| *1B Model Pretrained on 20B Tokens* | | | | | | |
| Social IQA | 0.387 | 0.411 | 0.406 | 0.406 | 0.394 | 0.401 |
| HellaSwag | 0.375 | 0.427 | 0.431 | 0.436 | 0.410 | 0.410 |
| PiQA | 0.658 | 0.684 | 0.693 | 0.691 | 0.684 | 0.680 |
| OpenBookQA | 0.278 | 0.298 | 0.300 | 0.304 | 0.299 | 0.302 |
| Lambada | 0.301 | 0.359 | 0.348 | 0.353 | 0.334 | 0.339 |
| SciQ | 0.802 | 0.822 | 0.809 | 0.828 | 0.821 | 0.818 |
| ARC Easy | 0.482 | 0.508 | 0.512 | 0.518 | 0.500 | 0.499 |
| COPA | 0.683 | 0.692 | 0.713 | 0.708 | 0.678 | 0.698 |
| RACE | 0.306 | 0.319 | 0.313 | 0.314 | 0.305 | 0.300 |
| LogiQA | 0.259 | 0.258 | 0.269 | 0.272 | 0.268 | 0.267 |
| WinoGrande | 0.513 | 0.527 | 0.541 | 0.512 | 0.535 | 0.533 |
| MultiRC | 0.523 | 0.504 | 0.510 | 0.530 | 0.529 | 0.548 |
| Average Performance | 0.464 | 0.484 | 0.487 | 0.489 | 0.480 | 0.483 |
| Estimated FLOPs | 0 | $3.7 \times 10^{19}$ (100%) | 0 | $3.5 \times 10^{18}$ (9.46%) | $9.66 \times 10^{16}$ (0.26%) | $9.66 \times 10^{16}$ (0.26%) |

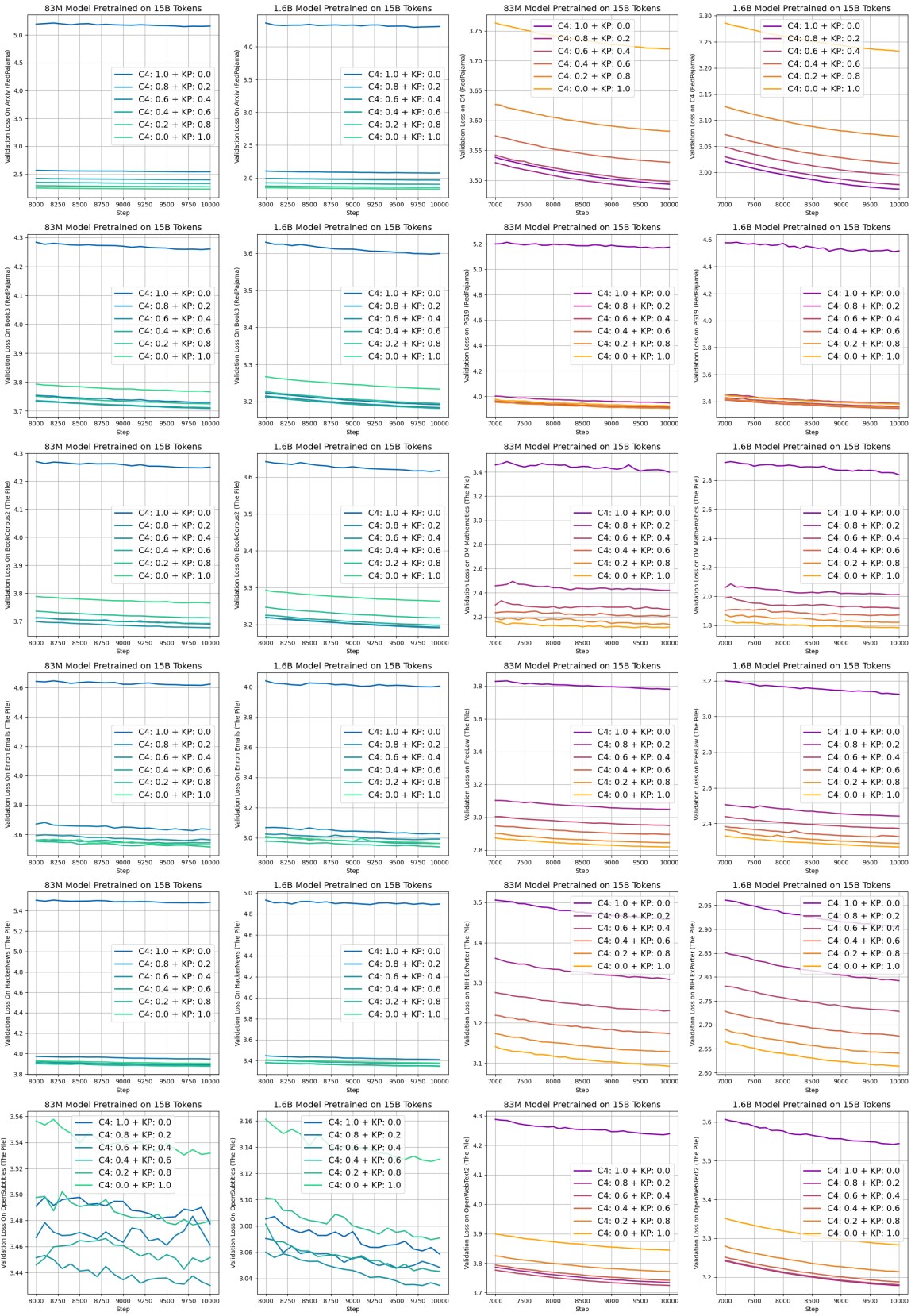

*Figure 8.* The validation loss on different dataset of models trained using data mixture in Table 1.

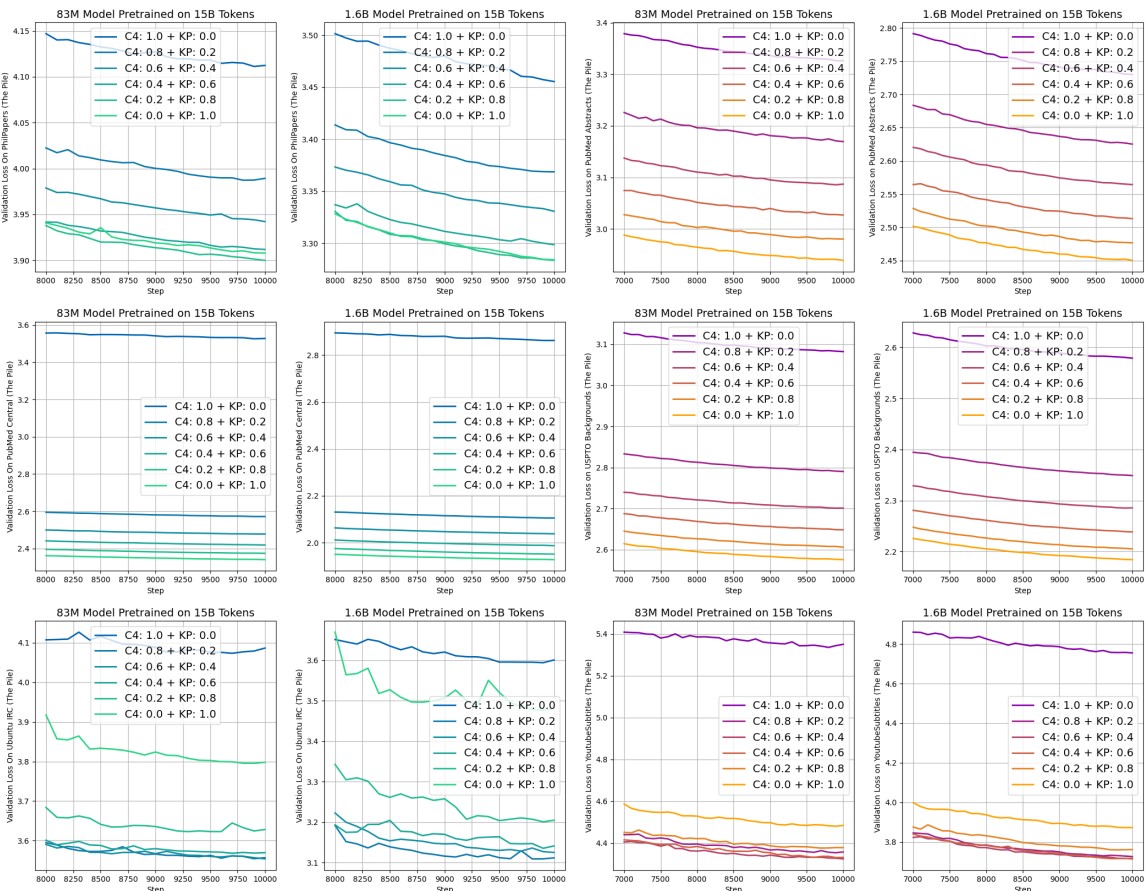

*Figure 9.* The validation loss on different dataset of models trained using data mixture in Table 1.

