# OpenReview forum: "Domain2Vec: Vectorizing Datasets to Find the Optimal Data Mixture without Training"
_ICML.cc/2025/Conference — ICML 2025 poster_

### Official Review · Reviewer_75ct · 2025-03-14

**Overall Recommendation:** 4

**Summary:**

The document introduces DOMAIN2VEC, a technique for optimizing data mixtures in training large language models by decomposing datasets into linear combinations of "Meta-Domains" to enable efficient identification of optimal data mixture ratios.
DOMAIN2VEC uses a meta-Domain classifier to classify any dataset and the Distribution Alignment Assumption (DA2) which suggests that the validation loss is low if the training and validation set are more aligned.

**Claims And Evidence:**

The authors have employed a K-means clustering resulting in 240 different Meta-Domain clusters for English and Chinese Data. Similarly, for code data, they classified code using 20 classes based on their programming language. They claim that they can decompose datasets according to these 260 Meta-Domains. However, it is unclear, especially for the English and Chinese text if this number of clusters is the appropriate one and the reason they chose K=240. Also, it is unclear whether these clusters which resulted from a specific dataset, can represent well other datasets.

**Essential References Not Discussed:**

To the best of my knowledge, there are no essential references that are not discussed.

**Experimental Designs Or Analyses:**

The experimental design seems valid. However, according to Table 3, the experimental results do not seem to significantly improve performance using DOMAIN2VEC.

**Methods And Evaluation Criteria:**

The authors use KNN as an embedding-based baseline without providing further details on that. The number of Nearest Neighbours K can have a significant effect on the data classification. Moreover, it seems that KNN performs better than DOMAIN2VEC + RegMix according to table 1.

**Other Comments Or Suggestions:**

No other comments.

**Other Strengths And Weaknesses:**

Strengths
The paper is well-written and contains extensive experimental results.
It touches on an important topic related to training Large Language Models.
It seems to improve computational cost with respect to previous state-of-the-art.

Weaknesses:
The experimental results do not demonstrate significant performance improvement.

**Questions For Authors:**

Why did you use K-means for finding the domains? Why did you choose to represent these domains with 240 Meta-domains?

**Relation To Broader Scientific Literature:**

Previous works use a proxy model or require resampling of the data mixtures.

**Theoretical Claims:**

All theoretical claims have been checked and seem valid.

---

> ### Author Rebuttal · Authors · 2025-04-01
>
> Dear Reviewer 75ct,
>
> Thanks for your very valuable review and recognition of our work! We will address your questions point by point.
>
> ## Q1 :Why did you use K-means for finding the domains? Why did you choose to represent these domains with 240 Meta-domains?   Also, it is unclear whether these clusters which resulted from a specific dataset, can represent well other datasets.
>
> A1: First, the 5.2TB text data we used to construct the Meta-Domains is unlabeled. We assume there exist distinct characteristics among different Meta-Domains, such as semantic features. In our implementation, we computed the embedding representations of these 5.2TB data to construct the Meta-Domains. Utilizing K-Means clustering on embeddings is an efficient approach under unsupervised conditions.
>
> Second, we referred to the Elbow method, selecting the number of Meta-Domains based on the point where inertia changes relatively gradually, as shown in Figure 1. Meanwhile, the chosen number of Meta-Domains in this paper is merely an experimental setting.
>
> Finally, the meta-domain classifier achieved an accuracy of 74.73% on the validation set. Given that this is a 260-class classification task, we believe that setting K = 240 effectively ensures clear distinctions among different clusters.
>
> ## Q2: The authors use KNN as an embedding-based baseline without providing further details on that. The number of Nearest Neighbours K can have a significant effect on the data classification. Moreover, it seems that KNN performs better than DOMAIN2VEC + RegMix according to table 2.
>
> A2: The details on the KNN baseline are as follows:
>
> ```
> First, for the training and validation datasets in section 4.1, we sampled 1000 examples from each dataset.
>
> Then, we used bge-small-v1.5 (since the datasets in section 4.1 are in English) to obtain embeddings for samples from each dataset and used mean pooling to get unique embeddings for each dataset.
>
> Meanwhile, we also used bge-small-v1.5 to obtain embeddings for the data in each Meta-Domain.
>
> Then, we set K as 1000 and used KNN (based on Euclidean distance) to obtain probability distributions of training and test datasets belonging to each Meta-Domain.
>
> Last, we treated these probability distributions as new domain vectors. Based on these domain vectors, we implemented the Distribution Alignment Assumption.
> ```
>
> Second, we believe that the superior performance of kNN validates the rationality of our Meta-Domain construction process. **However, it should be noted that DOMAIN2VEC + DA² still significantly outperforms KNN**.
>
> Last, we would like to clarify that: Embedding-based methods, in addition to having more limited context length compared to our method, still have several disadvantages. For different types of data, we used different clustering methods to construct Meta-Domains.
>
> 1. For code, we directly identified its programming language without using an embedding model.
>
> 2. For Chinese and English data, we used embedding models that output embeddings of different dimensions. Moreover, the semantic meaning of the same dimensions of different models' embeddings is obviously different.
>
> Therefore, our proposed method has better generalizability。
>
> ## Q3: However, according to Table 3, the experimental results do not seem to significantly improve performance using DOMAIN2VEC.
>
> A3: We want to clarify that Domain2Vec could provide a universal representation of pre-training datasets, which focuses on the scalability and efficiency of pre-training data mixture experiments.
>
> The scalability of Domain2Vec is reflected in: Domain2Vec establishes a latent space representation of datasets. Therefore, any pre-training data can be mapped into this space.
>
> Experiments conducted in the latent space remain consistent regardless of changes in pre-training datasets. In contrast, RegMix performs experiments at the dataset level, requiring all previous experimental results to be discarded and new experiments to be conducted when datasets change (such as, adding new datasets, improving the quality of some datasets).

---

### Official Review · Reviewer_H5YS · 2025-03-16

**Overall Recommendation:** 2

**Summary:**

This paper presents a method for determining the optimal data mixture weights for combining different pre-training datasets to train language models. The authors formulate this as an optimization problem, where the goal is to find the appropriate weights over a set of meta-domains. These meta-domains are constructed by applying K-means clustering to dataset embeddings. A meta-domain classifier is then trained to predict the probability of a dataset belonging to each meta-domain.
By representing both the training datasets and validation datasets as a linear combination of these meta-domains, the authors ensure that all datasets exist within the same representational space. This enables them to optimize the dataset mixture weights to minimize validation loss effectively.
Through extensive experiments, the authors demonstrate that their approach achieves performance comparable to prior methods such as DoReMi and RegMix while operating at a significantly lower computational cost.

**Claims And Evidence:**

Yes

**Essential References Not Discussed:**

There are a couple, and I note them in the weaknesses sections

**Experimental Designs Or Analyses:**

Yes

**Methods And Evaluation Criteria:**

Yes

**Other Comments Or Suggestions:**

NA

**Other Strengths And Weaknesses:**

Strengths:

The paper is easy to follow and the idea to breakdown dataset into a set of “building blocks” called meta-domains is interesting. The results are also impressive.

The paper should provide a more thorough discussion of its connections to prior related work. Specifically, Task2Vec (https://arxiv.org/pdf/1902.03545) introduces a framework for representing relationships between tasks, which conceptually aligns with the current study. The fundamental principle of meta-learning relies on identifying similar tasks and leveraging that information for training. Consequently, numerous studies have explored this, including those utilizing gradient similarity, such as the work presented in https://arxiv.org/pdf/1911.10600.
Moreover, dataset classification and dataset biases are well-recognized as challenging problems. The authors should elaborate on how their work relates to recent studies tackling these issues, such as the one presented in https://arxiv.org/pdf/2403.08632. Given these papers it is crucial to situate the current findings within the broader research landscape and highlight their contributions in relation to these prior efforts.
One more aspect that requires clarification is the discrepancy between the feature extraction model and the meta-domain classifier. The paper states that the features used for clustering are derived from the "bge-small" model, whereas the meta-domain classifier is trained using Qwen. The rationale behind this decision is not immediately apparent, and the authors should justify why these different models were chosen for these respective tasks. A clear explanation would strengthen the coherence of the methodological choices.
Another issue pertains to the clustering methodology. The K-means clustering approach used in the paper does not inherently ensure that clusters are not sufficiently independent as also observed in author’s experiments. To mitigate this, it would have been beneficial to introduce a diversity-enhancing objective within K-means and this could be implemented using Faiss,

A more critical concern relates to the use of linear regression on probabilities, as presented in Equation (6). Probabilities do not exist in a regular Euclidean space but instead lie on a simplex, making standard linear regression an inappropriate model choice. A more suitable approach would be geodesic regression and I would like the authors to comment on this.
Additionally, there appears to be a discrepancy in the reported calculations in line 377, where the paper states that "Pile-CC only shows a 4.01% improvement over Human." However, based on the values presented in Table 3, the correct computation appears to be: [(0.439-0.424)/0.439]*100=3.5%
rather than 4.01%. While I did not verify all numerical claims, I strongly recommend that the authors carefully re-evaluate their calculations to ensure accuracy. If my computation is incorrect, clarification would be helpful.
Overall, while the paper introduces interesting ideas, the identified technical inconsistencies significantly weaken its contributions. Addressing these concerns would considerably improve the rigor and credibility of the study. In its current form, I am unable to recommend acceptance.

**Questions For Authors:**

Please see weaknesses

**Relation To Broader Scientific Literature:**

This paper's contribution is relevant to the scientific literature in NLP but lacks discussion on other modalities

**Theoretical Claims:**

Not applicable

---

> ### Author Rebuttal · Authors · 2025-04-01
>
> Dear Reviewer H5YS,
>
> Thanks for your insightful review and suggestions! We will respond to your questions one by one.
>
> ## Q1: The paper should provide a more thorough discussion of its connections to prior related work, i.e, Task2Vec [1].
>
> A1: Similar to [2] and [3] cited in lines 407–412 (right column), Task2Vec [1] is an efficient way to represent a task or its corresponding dataset as a fixed-dimensional vector.
>
> However, Domain2Vec differs from these works in both purpose and implementation, as we focus on data mixture for language model pretraining rather than using a method like Task2Vec to select an expert from a collection (which can improve test performance while adding only minimal overhead to the training process).
>
> Last, we will also include a citation for Task2Vec.
>
> [1] https://arxiv.org/pdf/1902.03545
>
> [2] https://arxiv.org/abs/1905.11063
>
> [3] https://arxiv.org/abs/2406.00281
>
> ## Q2: Dataset classification and dataset biases are well-recognized as challenging problems. The authors should elaborate on how their work relates to recent studies tackling these issues, such as the one presented in [4].
>
> A2: We would like to clarify that the target of Domain2Vec is to determine which Meta-Domains a given dataset can be composed of.
>
> In practice, when applying Domain2Vec, we are actually performing a text classification task rather than classifying an entire dataset. As shown in Figure 6, different datasets can share the same Meta-Domain, which explains why they mutually benefit from training with each other.
>
> Last, We will also add references to [4] and discuss it in future work.
>
> [4] https://arxiv.org/pdf/2403.08632.
>
> ## Q3: The paper states that the features used for clustering are derived from the "bge-small" model, whereas the meta-domain classifier is trained using Qwen. The rationale behind this decision is not immediately apparent, and the authors should justify why these different models were chosen for these respective tasks.
>
> A3: There are few reasons we choose Qwen as the backbone rather than the bge model.
>
> 1) Since the embedding dimensions and semantic feature spaces of the Chinese and English bge models are inconsistent, and no embedding model was used for code data.
>
> 2) The Embedding models like bge have a context window limited to 512 tokens, while pre-training data is typically longer than 512 tokens. In contrast, our Meta-Domain Classifier, trained based on Qwen, can handle context lengths of up to 8k tokens or even longer.
>
> 3) More importantly, our Meta-Domain classifier can output very specific probability distribution on the meta-domains while the baseline can only output a hard assignment (0 or 1). There are indeed some kNN algorithms that can output soft scores, but the score is indirectly based on the distance to the center of clusters. The efficiency of kNN is also limited because it is a lazy learning algorithm and puts time complexity into inference time.
>
> ## Q4：The K-means clustering approach used in the paper does not inherently ensure that clusters are not sufficiently independent as also observed in author’s experiments. To mitigate this, it would have been beneficial to introduce a diversity-enhancing objective within K-means and this could be implemented using Faiss.
>
> A4: First, this is a 260-class classification task, and our meta-domain classifier achieved an accuracy of 74.73% on the validation set. Thus, we believe that the K-means clustering approach effectively ensures clear distinctions among clusters.
>
> Second, we have utilized FAISS to perform K-means clustering. In the future, we plan to explore introducing a diversity-enhancing objective within the K-means clustering process.
>
> ## Q5: A more critical concern relates to the use of linear regression on probabilities, as presented in Equation (6). Probabilities do not exist in a regular Euclidean space but instead lie on a simplex, making standard linear regression an inappropriate model choice. A more suitable approach would be geodesic regression and I would like the authors to comment on this.
>
> A6: Great question! We want to clarify that the domain vector is not merely the probabilities assigned by the Meta-Domain Classifier to a dataset. Its deeper implication is that once we obtain the domain vector of a dataset, the dataset can be regarded as a **linear combination** of various Meta-Domains represented by the domain vector. Therefore, we can apply algorithms such as RegMix (Equation (6)) on Domain2Vec. In future work, we also plan to explore methods like geodesic regression.
>
> ## Q7: While I did not verify all numerical claims, I strongly recommend that the authors carefully re-evaluate their calculations to ensure accuracy.
>
> A7: Thank you for your careful review. This is indeed a typo error. We have also verified other numerical claims throughout the paper, and this particular typo does not affect the conclusions presented.

---

### Official Review · Reviewer_HtfL · 2025-03-19

**Overall Recommendation:** 1

**Summary:**

Authors propose a sampling method for training LLMs on multiple sources. Their core idea is as follows: They construct a universal set of real-valued vectors from a large textual corpus--using K-means and doc embeddings. Each vector approximately represents a topical domain of the corpus. Then they take random samples from the training sources of LLM and use a classifier to determine on average which vectors are most similar to the sampled documents. They use these similarity scores to represent the entire multi-source training dataset. They follow the same procedure for the validation set as well. Then they learn a coefficient set that transforms the representation of the training set into the validation set. The obtained weight set is the desired sampling ratio.

Conceptually, their idea is similar to what is used in topic-modeling, LSI, or LDA, but the vector entries here are words. So they extract a set of vectors from train and validation sets that best represent these two sets. Then they try to learn a set of coefficients that makes the train matrix similar to the validation matrix. Their argument is that if the model works on validation set, then it will work on unseen sets (the real test set).

**Claims And Evidence:**

Please see the section below

**Essential References Not Discussed:**

N/A

**Experimental Designs Or Analyses:**

Most of them

**Methods And Evaluation Criteria:**

Yes

**Other Comments Or Suggestions:**

None

**Other Strengths And Weaknesses:**

**Strengths:**

-  The topic is timely.
-  The method is intuitive.
-  Then analysis is convincing.

**Weaknesses:**

-  Reading Section 2 (which is the core part of the paper) was very painful. I had to read it at least three times, and for some parts of it even more. Here are some of the issues:

   - Line 88 (right column). what does this mean: (where each element vj of v represents the projection (weight) of the dataset D on Dj\*). what does it mean to have a weight of a dataset on Dj\*? After the Key assumption please explain what the reader is expecting to see. You suddenly jump into explaining that you are collecting data, the reader would wonder what you would need the data for. Line 136 (left column), how did you train the classifier, what is the training data? Please re-organize the section and put the training explanation before saying how you would use the classifier. Line 143 (left col), what is "domain vector"? is it something that represents a document or a dataset? On Line 143 you are using it for a doc, but on line 82 (right col) you are using it for a dataset. Line 156 (left com), when you define V_train the inner vectors should be transposed. Lines 159 and 162 (left col), why is there meta-domain once capitalized and once is not? is there ant difference between the two? Line 160 (left col), what does it mean when you say a "text belongs to a meta-domain". As far as I know the verb "belong" is used in the set theory and it means membership.

   As you can see the explanations are very vague, the organization of the section is messy, and  the wording is not scientific.

-  The main reason that I oppose accepting this paper is this: the core idea of the authors to do the sampling such that the sampled documents from the training set become similar to the sampled documents from the validation set is a text-book example of overfitting. The role of validation set is to only validate your ML model, not to use it in the learning algorithm. The goal of ML is to develop a model to have a low error rate in an UNSEEN set, and the validation set can be used as an unseen set during the learning stage. But you are using the validation set inside the learning model itself. This is why your improvement is almost zero, and the algorithm is not generalizable.

**Questions For Authors:**

None

**Relation To Broader Scientific Literature:**

N/A

**Theoretical Claims:**

Yes

---

> ### Author Rebuttal · Authors · 2025-04-01
>
> Deal Reviewer HtfL,
>
> Thanks for your careful review! We will reply to your questions one by one and hope to solve your concern.
>
> ## Part1: Clarification of Our Method
>
> ### Q1: Line 88 (right column). what does this mean: (where each element vj of v represents the projection (weight) of the dataset D on Dj*). what does it mean to have a weight of a dataset on Dj*?
>
> A1: First, our proposed Domain2Vec can transform a dataset into a domain vector $v$. Next, we propose that any dataset can be viewed as a **linear combination** of the domain vectors for multiple Meta Domains $\mathcal D_j^*$. And the weight of this linear combination is $v$. Therefore, each element $v_j$ of $v$ represents the projection (weight) of the dataset $\mathcal D$ on  $\mathcal D_j^*$.
>
>
> ### Q2: Line 143 (left col), what is "domain vector"? is it something that represents a document or a dataset?
>
> A2: First, we treat the domain vector as a vector representation of datasets.
>
> For a given dataset, we first sample N documents from it.
>
> For each document $text_j$, $\mathcal{p}_i$ represents the probability that $text_j$ originates from the i-th Meta-Domain.
> It should be noted that in our paper, for each document $text_j$, $\mathcal{p}_i$ is also referred to as the domain vector, since a single text can be viewed as a dataset with a sample size of 1.
>
> Therefore, the domain vector of the given dataset is obtained by averaging the domain vectors of documents sampled from it.
>
> ### Q3: The role of validation set is to only validate your ML model, not to use it in the learning algorithm.
>
> A3: First, **in previous studies on data mixture [1][2][3][4], it is common practice to use a validation set to optimize the data mixture, which should not be regarded as overfitting.** For example, Data Mixing Law, RegMix, D-CPT-Law, and BiMix all model the relationship between data mixture and validation loss to identify the optimal mixture that minimizes validation loss.
>
> Second, the idea of distribution matching has also appeared in previous works[5]. For instance, [5] improves the performance of continued pretraining by selecting a subset of a large raw unlabeled dataset to match a desired target distribution given unlabeled target samples.
>
> Third, our testing is independent of the validation set.
> 1) Section 4.1 only evaluates whether Domain2Vec can accurately predict the model’s validation loss.
> 2) When evaluating downstream task performance, we use downstream tasks that are entirely independent of the validation set, and our experimental setup follows the approach of RegMix[2].
>
> Last, compared with other baselines, our method significantly reduces computational overhead. Moreover, the data mixture obtained by our method achieves clear performance improvements on downstream tasks compared to the original mixture of The Pile dataset. Therefore, the improvement brought by our method is not "almost zero."
>
> [1] https://openreview.net/pdf?id=jjCB27TMK3
>
> [2] https://openreview.net/forum?id=5BjQOUXq7i
>
> [3] https://openreview.net/pdf?id=JzKFN5fWOk
>
> [4] https://openreview.net/forum?id=JsM46OZix7
>
> [5] https://arxiv.org/pdf/2302.03169
>
>
> ## Part2: Comments Suggestions And Typos
>
> ### Q4: Line 136 (left column), how did you train the classifier, what is the training data?
>
> A4: First, in lines 158 (left column) through 125 (right column), we have provided a detailed explanation of the meta domain classifier training. Next, we will swap the order of the sections in lines 141–157 (left column) with those in lines 158 (left column) through 125 (right column) to make this part clearer.
>
> ### Q5: After the Key assumption please explain what the reader is expecting to see. You suddenly jump into explaining that you are collecting data, the reader would wonder what you would need the data for.
>
> A5: In line 91 (right column), we will add a corresponding transitional sentence for better clarity.
>
> ### Q6: Line 156 (left com), when you define V_train the inner vectors should be transposed.
>
> A6: Thanks for your suggestions. We will fix this typo error.
>
> ### Q7: Lines 159 and 162 (left col), why is there meta-domain once capitalized and once is not? is there and difference between the two?
>
> A7: There is no difference here. We will capitalize "meta-domain" in line 162.
>
> ### Q8: Line 160 (left col), what does it mean when you say a "text belongs to a meta-domain". As far as I know the verb "belong" is used in the set theory and it means membership.
>
> A8: Perhaps "originate from" would be a better expression. We will carefully review the usage of certain phrases.

---

### Official Review · Reviewer_ZTKc · 2025-03-25

**Overall Recommendation:** 3

**Summary:**

Domain2Vec introduces a method for vectorizing datasets by decomposing them into linear combinations of Meta-Domains, which enables efficient identification of optimal dataset mixtures for LLM pretraining. They sampled and embedded texts from which predetermined clusters/labels, which they then clustered and trained the labels on with a classifier head.
Domain2Vec matches dataset distributions by minimizing validation loss differences without the heavy computational cost of training proxy LLMs.
The method achieves comparable dataset mixture quality to existing approaches like DoReMi and RegMix at significantly reduced computational expense (only 0.26% of DoReMi’s cost).

**Claims And Evidence:**

The paper claims substantial computational savings compared to baseline methods (DoReMi and RegMix), with evidence supported by experiments demonstrating Domain2Vec's ability to find optimal data mixtures.

However, there are two unaddressed claims:
- DoReMi demonstrated its efficacy on two datasets (The Pile and The Glam), while this paper only focused on The Pile. Domain2Vec involves much more modelling primitives (an embedding model, training a classifier head, etc) and other hyperparameters, as such, demonstrating the method’s robustness is necessary.
- Specific claim of DoReMi requiring 3.7e19 FLOPS is not clearly substantiated within the reviewed paper or the original DoReMi work.

**Essential References Not Discussed:**

Distribution Alignment Assumption (or target distribution matching) are shown in prior works but not cited. It’ll be interesting to see simpler methods like the Hashed NGram method in Xie et al [1] compared as well.

[1] Sang Michael Xie and Shibani Santurkar and Tengyu Ma and Percy Liang. Data Selection for Language Models via Importance Resampling. https://arxiv.org/abs/2302.03169

[2] Suchin Gururangan and Ana Marasović and Swabha Swayamdipta and Kyle Lo and Iz Beltagy and Doug Downey and Noah A. Smith. Don't Stop Pretraining: Adapt Language Models to Domains and Tasks. https://arxiv.org/abs/2004.10964

**Experimental Designs Or Analyses:**

The paper conducts extensive experiments comparing Domain2Vec against established baselines, showing that Domain2Vec achieves similar downstream task performance at dramatically lower computational cost.
The baselines of DoReMi and RegMix are well thought and carried out.
However, additional analyses on dataset hyperparameter sensitivity (e.g., varying the number of Meta-Domains and how that would affect the classifier and the end performance) are needed.

**Methods And Evaluation Criteria:**

Domain2Vec's method involves embedding datasets into a "Meta-Domain" space, clustering embeddings, and training a classifier head on top of a small LM to produce domain vectors. Evaluation is performed primarily on the Pile dataset, comparing Domain2Vec’s performance against baselines through validation loss and downstream task accuracy. The evaluation is thorough and well-done, but limited to a single dataset (The Pile).

**Other Comments Or Suggestions:**

Minor presentation improvements, i.e. enlarging plot axes and legends for clarity (Figures 3, 4, 5, and 6).

**Other Strengths And Weaknesses:**

Strengths:

Domain2Vec significantly reduces computational overhead; strong comparative experiments; clear methodological explanations. I believe it would be a helpful addition to the data curation/mixture/filtering stack, given the transferability of the classifier and the simplicity of the methodology.

Weaknesses:

The complexity of the embedding and classification steps could benefit from additional robustness demonstrations. Limited hyperparameter ablation (260 meta domains). Validation was performed primarily on one dataset.

**Questions For Authors:**

1. Could you provide explicit calculations or further clarify the claim regarding DoReMi's and RegMix’s computational cost (3.7e19 FLOPS)?

**Relation To Broader Scientific Literature:**

The current established works on Data Mixtures are DoReMi, RegMix, Data Mixture Laws (Ye et al). All of these papers require training and using smaller proxy models to inform data mixture for larger models. Domain2Vec effectively bypasses the need to train any small proxy model, by leveraging an embedding model instead. Thus enabling computational savings.
Furthermore, all the above works do not generalize to new datasets, as they’ll require retraining of the small proxy models, while Domain2Vec works out of the box with its trained classifier. As such, if proven to be robust, this work has a meaningful impact on the field of pretraining data mixtures.

**Theoretical Claims:**

None

---

> ### Author Rebuttal · Authors · 2025-04-01
>
> Dear Revierwer ZTKc,
>
> Thank you for your recognition of the value of our work, as well as for your valuable comments.  We will respond to your questions one by one.
>
> ## Q1: Domain2Vec involves much more modelling primitivesand other hyperparameters, as such, demonstrating the method’s robustness is necessary.
>
> A1: In section 4.1, we use a mixture of C4 and the Knowledge Pile as the training set, and the Pile and RedPajama as the validation set. Experiments on different datasets (Sec 4.1: C4, Knowledge Pile, Sec 4.2: The Pile) demonstrate the robustness of Domain2Vec.
>
> ## Q2: Specific claim of DoReMi requiring 3.7e19 FLOPS is not clearly substantiated within the reviewed paper or the original DoReMi work. Could you provide explicit calculations or further clarify the claim regarding DoReMi's and RegMix’s computational cost?
>
> A2: Of course, the estimated FLOPS of various baslines are from the Tabel 4 of RegMix paper.
>
> [1] https://arxiv.org/abs/2407.01492
>
> ## Q3: However, additional analyses on dataset hyperparameter sensitivity (e.g., varying the number of Meta-Domains) are needed.
>
> A3: Theoretically, increasing the number of Meta-Domains will lead to more accurate Domain2Vec representations of pretraining datasets. Due to limited computational resources, we only experimented with the number of Meta-Domains during clustering (Figure 3). Exploring how varying the number of Meta-Domains affects both the classifier and the final performance is left for our future work.
>
> ## Q4:  Distribution Alignment Assumption (or target distribution matching) are shown in prior works but not cited. It’ll be interesting to see simpler methods like the Hashed NGram method in Xie et al [2] compared as well.
>
> A4: First, we will cite [2] in our next version.
>
> Second, we would like to clarify that although [2] and our proposed Distribution Alignment Assumption share certain similarities:
>
> 1) The methods used for feature construction differ. [2] is based on Hashed N-gram Features, whereas Domain2Vec uses a Meta-Domain Classifier to generate Domain Vectors.
>
> 2) [2] conducts data selection at the sample level, whereas Domain2Vec performs data mixing at the dataset level across different datasets.
>
> 3) [2] is validated on encoder-only language models such as BERT and RoBERTa, while Domain2Vec is validated on autoregressive, decoder-only language models.
>
> [2] https://arxiv.org/pdf/2302.03169
>
> ## Q5: Minor presentation improvements, i.e. enlarging plot axes and legends for clarity (Figures 3, 4, 5, and 6).
>
> A5: Thanks for your suggestions, and we will increase the font size in all the figures.

---

### Decision · Program_Chairs · 2025-05-01

**Decision:**

Accept (poster)

**Comment:**

This paper proposes an approach for efficiently identifying optimal dataset mixtures for LLM pretraining by vectorizing datasets through decomposition into meta-domains. This method achieves comparable dataset mixture quality to existing approaches like DoReMi and RegMix but with a significantly reduced computational expense. With more rigorous analysis, this work can add to the literature on understanding datasets.

A significant concern regarding the methodology revolves around the use of the validation set in the learning algorithm for distribution matching. Reviewer HtfL strongly argues that using the validation set to influence the training process by aligning its distribution with the training set is a textbook example of overfitting. It's worth noting that compared to traditional ML, here the validation set's domain vector is being used as a target to align the training data's domain vector. I do not view this issue major enough to warrant rejection, especially given the widespread use of val set reuse and prior empirical evidence that it does not lead to overfitting (e.g., ImageNet V2). However, given the concerns raised, I urge the authors to provide a thorough analysis of potential overfitting effects.